# Carbon, nitrogen, and phosphorus stoichiometry of organic matter in Swedish forest soils and its relationship with climate, tree species, and soil texture

Marie Spohn[1], Johan Stendahl[1]

[1]Department of Soil and Environment, Swedish University of Agricultural Sciences (SLU), Lennart Hjelms väg 9, P.O. Box 7014, 75007 Uppsala, Sweden

*Correspondence to*: Marie Spohn (marie.spohn@slu.se)

**Abstract.** While the carbon (C) content of temperate and boreal forest soils is relatively well studied, much less is known about the ratios of C, nitrogen (N), and phosphorus (P) of the soil organic matter, and the abiotic and biotic factors that shape them. Therefore, the aim of this study was to explore carbon, nitrogen, and organic phosphorus (OP) contents and element ratios in temperate and boreal forest soils and their relationships with climate, dominant tree species, and soil texture. For this

purpose, we studied 309 forest soils located all over Sweden between 56°N and 68°N. The soils are a representative subsample of Swedish forest soils with a stand age >60 years that were sampled for the Swedish Forest Soil Inventory. We found that the N stock of the organic layer increased by a factor of 7.5 from -2.0°C to 7.5°C mean annual temperature (MAT), which is almost twice as much as the increase of the organic layer stock along the MAT gradient. The increase in the N stock went along with an increase in the N:P ratio of the organic layer by a factor of 2.1 from -2.0°C to 7.5°C MAT ($R^2$=0.36, p<0.001).

Forests dominated by pine had higher C:N ratios in the organic layer and mineral soil down to a depth of 65 cm than forests dominated by spruce. Further, also the C:P ratio was increased in the pine-dominated forests compared to forests dominated by other tree species in the organic layer, but the C:OP ratio in the mineral soil was not elevated in pine forests. C, N and OP contents in the mineral soil were higher in fine-textured soils than in coarse-textured soils by a factor of 2.3, 3.5, and 4.6, respectively. Thus, the effect of texture was stronger on OP than on N and C, likely because OP adsorbs very rigidly to mineral

surfaces. Further, we found that the P and K concentrations of the organic layer were inversely related with the organic layer stock, while the N:P ratio was positively related to the organic layer stock. Taken together, the results show that the N:P ratio of the organic layer was most strongly related to MAT. Further, the C:N ratio was most strongly related to dominant tree species, even in the mineral subsoil. In contrast, the C:P ratio was only affected by dominant tree species in the organic layer, but the C:OP ratio in the mineral soil was hardly affected by tree species due to the strong effect of soil texture on the OP

concentration.

## 1. Introduction

Temperate and boreal forests store large amounts of soil organic matter (SOM) (Bradshaw and Warkentin, 2015; Tau-Strand et al., 2016; Hounkpatin et al., 2021). While most studies on SOM in temperate and boreal forests concentrated exclusively on carbon (C), and some studies also included nitrogen (N) and investigated the C:N ratio (Callesen et al., 2007; Cools et al., 2014; Tau-Strand et al., 2016), very few studies explored the phosphorus (P) content, and the C:P and N:P ratios of SOM in forests (for an exception see Tipping et al., 2016), particularly in North European forests. Thus, we lack knowledge about the

C:N:P stoichiometry of the organic matter of temperate and boreal forest soils and the factors that control it.

The N stock of well-drained temperate and boreal forest soils in Northern Europe increases with increasing mean annual temperature (MAT) (Callesen et al., 2007). Yet, it is unknown whether organic P (OP) increases in the same way as N along the MAT gradient in Northern Europe or if the N:P ratio of SOM changes with MAT in North European forest soils. If P availability is high enough to balance the elevated N availability at sites with high N availability and high MAT, the N:P ratio

of SOM in North European forests will likely not change with MAT. However, several studies question whether P availability is sufficient to balance the high rates at which N is incorporated into plant biomass in N-rich European forests (Flückiger and Braun, 1998; Braun et al., 2010; Talkner et al., 2015; Jonard et al., 2015; Heuck et al., 2018). In particular, it has been suggested based on modeling exercises that high N availability in South Sweden might lead to P limitations in trees (indicated by molar needle N:P ratio >12) in South Swedish forests (Akselsson et al., 2008; Yu et al., 2018), which likely translates into high N:P

ratios of plant litter and potentially of SOM (Zechmeister-Boltenstern et al., 2015). Yet, on the other hand, it has to be considered that most soils in Scandinavia are only between 9,000 and 14,000 years old. Thus, comparatively little P leaching and P occlusion has likely occurred during the short duration of pedogenesis, resulting probably in a relatively high P availability in soils of all climate regimes, which should be able to balance the high N availability at sites with high MAT. Taken together, it is currently still an open question if and how the N:P ratio of SOM in Swedish forest soils changes with

organic matter N stock and MAT.

Several studies showed that the C:N ratios of the organic layer and the mineral topsoil of forests in North Europe are affect by the dominant tree species (Vesterdal and Raulund-Rasmussen, 1998; Vesterdal et al., 2008; Hansson et al. 2011; Cools et al. 2014). In particular, it has been reported that the organic layers of pine forests have a higher C:N ratio than spruce and deciduous forests (Vesterdal and Raulund-Rasmussen, 1998; Vesterdal et al., 2008; Hansson et al. 2011; Cools et al. 2014).

However, little is known about the C:N ratio in the mineral soil below the uppermost 10 cm. Furthermore, very little is known about the relationship between dominant tree species and the C:OP and N:OP ratio of the organic layer and the mineral soil in North European forests (for exceptions see Vesterdal and Raulund-Rasmussen, 1998; Hansson et al., 2011). Since N and P contents in leaf litter are strongly associated (McGroddy et al., 2004), it can by hypothesized that C:N and C:P ratios show similar differences between forests dominated by different tree species.

Besides climate and tree species, the SOM stoichiometry is likely also affected by soil texture. Recent research showed that OP is more strongly enriched in the clay-size fraction than organic C because OP compounds adsorb rigidly to mineral surfaces in soils (Spohn, 2020a). This suggests that OP has also higher concentrations in fine-textured than in coarse-textured soils because the adsorption of organic compounds protects them against microbial decomposition, and thus should lead to a preferential enrichment of the compounds that adsorb particularly strongly in soils that provide many sorption sites. Further,

it seems likely that besides OP, N is also enriched in fine-textured soils because a large part of N is present in SOM in the form of charged nitrate or ammonium groups that can adsorb to charged surfaces (Knicker et al., 1993; Jones and Hodges, 1999; Kopittke et al., 2018; Miltner et al., 2009). However, previous studies on the effect of texture on the soil N concentration and the C:N ratios in North European forest soils arrived at contradicting conclusions (Vejre et al., 2003; Callesen et al., 2007). The aim of this study was to explore the C, N, and OP contents and element ratios in temperate and boreal forest soils in

Sweden and their relationships with climate, dominant tree species, and soil texture. We hypothesized, first, that while the N stock of the organic layer increases with increasing MAT, the N:P ratio of the organic layer does not change with MAT. We hypothesized second, that the C:OP ratio of the organic layer and mineral soil shows similar differences between forests with different tree species as the C:N ratio. Third, we hypothesized that organic matter is N- and P-richer in fine-textured than in coarse-textured mineral soils. In order to test these hypotheses, we studied 309 forest soils with a forest stand age >60 years

located all over Sweden between 56°N and 68°N. The soils are a representative subsample of Swedish forest soils with a stand age >60 years that were sampled for the Swedish Forest Soil Inventory. We selected sites with a stand age >60 years in order to minimize the effect of forest management (such as clear-cutting) on the results.

## 2. Material and Methods

**2.1 Data collection, sampling, and sample preparation**

    Data and soil samples were collected for the Swedish Forest Soil Inventory (SFSI), which is conducted together with the National Forest Inventory (Riksskogstaxeringen) since 1923. The SFSI in its current form started in 1983 and monitors the state of the Swedish forests with respect to soil and vegetation. It covers all of Sweden except for arable land and urban areas. The inventory visits about 20,000 permanent plots over a 10-year period, sampling every year about 2,000 plots distributed all

over Sweden. Circular plots with 10 m radius are located in quadratic clusters on a triangular grid (Ranneby et al., 1987), which is denser towards the south of the country to account for the greater fragmentation of the landscape and a more diverse geology in the south of Sweden. Each quadratic cluster encompasses 8 circular plots (4 in the southwestern region; Ranneby et al., 1987). At each of these circular plots the dominant tree species was classified according to the following classes based on basal area: Deciduous (birch, aspen, beech, and oak), Mixed (which are mainly mixed coniferous forests), Pine (*Pinus*

*sylvestris* + *Pinus contorta*), and Spruce (*Picea abies*). The diameter at breast height is determined for all trees, and from the differences over 10 years, the stem growth rate of the trees at plot level is calculated. A loss of one or more tree(s) can therefore

result in a decline in standing volume between inventories, and thus a negative stem growth rate. In addition, the stand age is determined from the stand history assessed by the repetitive inventory together with wood coring in the 1980s.

Soil sampling is carried out on a subset of the circular plots, i.e. organic layer sampling on c. 10,000 plots and mineral soil sampling on c. 4,500 plots. The organic layer is sampled volumetrically using a 10 cm diameter corer in a 3.14 m$^2$ subplot within each circular plot throughout the entire depth of the organic layer (up to 30 cm depth), excluding the litter layer. To gain a sample of about 1.5 liter, 1-9 volumetric samples are combined. In addition, a small soil profile is prepared in the subplot. The soil order is determined according to WRB (FAO, 2013), and mineral soil is collected at fixed depth intervals: 0–

10 cm, 10–20 cm, and 55–65 cm. The texture of the mineral soil samples is determined in the field according to the following classes; clay, fine silt, coarse silt, fine sand, sand, and coarse sand. In the laboratory, all soil samples are dried to constant weight at 35°C. The samples are homogenized and sieved (<2 mm), and living and dead roots >1 mm diameter are removed from the mineral soil samples. The samples are weighed and the stock of the organic layer is calculated based on the weight of the fraction <2 mm of the organic layer. The organic layer stock is the mass of the organic layer on an area basis (t ha$^{-1}$).

Chemical analyses are carried out on the fine soil fraction (<2 mm).

For the present study, we selected plots that (i) have been sampled between 2013 and 2018, (ii) were covered by forest with a long-term productivity >1 m$^3$ yr$^{-1}$ and a stand age >60 years, (iii) had the humus form mor or moder (which excludes peatlands and plots with humus form mull), and (iv) for which data on the concentration of P in the parent material (in a depth of 50 cm) was available (Olsson et al., 1993). We excluded plots with the humus form mull because at these plots, the mull humus is

was available (Olsson et al., 1993). We excluded plots with the humus form mull because at these plots, the mull humus is classified as an A horizon and no organic layer is sampled for the SFSI. The plot selection based on these four criteria resulted in a total number of 309 plots, also called forests in the following (Fig. 1). Of these, 203 were podzols, 84 were regosols, 12 were arenosols, three were leptosols, three were gleysols, three were cambisols, and one soil was an umbrisol. Two of the three gleysols were covered by pine forest and the other one by spruce forest. The mean stand age was 113 years. Of the 309 sites,

119 sites had a stand age >120 years, and are thus generally classified as old growth forests (not planted). Further, 25 sites were on formally set-aside land. Only 10 forests were classified as deciduous with five of them being beech and oak forests.

**2.2 Soil chemical analyses and climate data**

Most soil chemical analyses were conducted for the Swedish Forest Soil Inventory based on the samples taken between 2013

and 2018. The total C and N content was analyzed using an elemental analyzer (TruMac CN, LECO). pH was determined in water (soil: water ratio 1:2.5) using a Pt electrode (Aquatrode plus Pt1000, Methrom). Exchangeable cations (Ca, Mg, Na, K, and Mn) were extracted in 1 M ammonium acetate buffered at pH 7.00 and analyzed by inductively coupled plasma optical emission spectroscopy (ICP-OES; Avivo 200, Perkin Elmer).

In addition to the variables determined for the Swedish Forest Soil Inventory, we measured total P and organic P for the present study based on the samples taken between 2013 and 2018. Total P of the organic layer and the mineral soil in a depth of 0-10 cm was extracted in nitric acid and hydrogen peroxide (in a ratio 1:3) in a microwave system (Ethos Easy, Milestone) according to the instructions of the manufacturer, and P was determined using ICP-OES (Avivo 200, Perkin Elmer) after filtration of the extract.

Organic P in the mineral soil in a depth of 0-10 cm was determined according to Saunders and Williams (1955) and Williams and Saunders (1956) as specified in Pansu and Gautheyrou (2006). Briefly, each sample was separated in two aliquots, each of 1 g. The first aliquot was directly extracted in 0.5 M $H_2SO_4$ on a horizontal shaker for 16 h. The other aliquot was ignited at 550°C for 2 h and subsequently extracted in $H_2SO_4$ in the same way as the non-ignited sample. Inorganic P was measured in the extracts by the molybdenum blue method according to Murphy and Riley (1962) using a continuous flow system (AA500, Seal). Total organic P was calculated as the difference between inorganic P in ignited and non-ignited samples.

Total C in this study is considered organic C since all soils were acidic. Furthermore, total P in the organic layer is considered to be organic P. For the mineral soil, we specify whether we refer to organic or total P, in the following.

Total P in the parent material was determined on samples that were collected at the plots of the SFSI from the B/C and C horizon in the 1980s. The samples were ground, sieved (< 2mm), ignited and fused with lithium-metaborate, and subsequently, total P was determined by ICP (Olsson et al., 1993).

Other data used in this study are mean annual temperature (MAT; from 2012), mean annual precipitation (MAP; from 2012) and total N deposition (NDep; from 1998) retrieved from the Swedish Meteorological and Hydrological Institute (https://www.smhi.se/data). We assume that the N deposition flux data of the year 1998 is representative for historic N deposition fluxes in Sweden during the last decades which were highest in the 1990s.

**2.3 Data analysis**

For the analysis of soil texture, the texture class clay and fine silt were combined as well as the texture class sand, and coarse sand. Organic layer element stocks were calculated by multiplying the organic layer stock with the respective element concentration. All element ratios were calculated on a molar basis (mol $mol^{-1}$). We calculated linear models, power functions, medians and arithmetic means, and conducted (multiple) regression analyses. For the regression analyses, all soil chemical variables (element stocks, element concentrations, and element ratios) as well as the organic layer stock and the P concentration in the parent material were transformed by calculating their natural logarithm (log-transformation) since they tended to be not normally distributed but were right-skewed. In addition, also the variable atmospheric N deposition was log-transform prior to regression analysis because it was not normally distributed. The variables latitude, MAT, and texture class were not transformed for the regression analysis. All significant regressions referred to in the text are significant at $p<0.001$ (if not indicated otherwise), and individual p values for specific regressions are given in the tables. Furthermore, we conducted

ANOVA followed by Tukey test after log-transformation of the data, and we considered p=0.05 as the threshold for significance. All data analyses were conducted using R (version 4.1.1, R Core Team, 2021).

## 3. Results

### 3.1 Latitude, mean annual temperature, N deposition, and SOM stoichiometry

We found that many soil properties were correlated with latitude, MAT, and N deposition, but hardly with MAP (Table 1). The log-transformed N stock of the organic layer was positively correlated with MAT ($R^2$=0.30; Table 1) and with the log-transformed N deposition ($R^2$=0.30), and negatively with latitude ($R^2$=0.26; Table 1). In the linear model fitted to the data, the N stock of the organic layer increased by a factor of 7.5 from 192 kg ha$^{-1}$ at -2.0°C MAT to 1439 kg ha$^{-1}$ at 7.5°C MAT (Fig. 2a).

The log-transformed organic layer stock was positively correlated with MAT ($R^2$=0.24; Table 1) and with the log-transformed N deposition ($R^2$=0.23), and negatively with latitude ($R^2$=0.20; Table 1). In the linear model fitted to the data, the organic layer stock increased by a factor of 3.9 from 30.7 t ha$^{-1}$ at -2.0°C MAT to 120.8 t ha$^{-1}$ at 7.5°C MAT (Fig. 2b).

Atmospheric N deposition was below 1.5 kg ha$^{-1}$ year$^{-1}$ in most forests, but in some, it was higher, and in 8 forests it was even between 5.5 and 8 kg ha$^{-1}$ year$^{-1}$ (Fig. 2c). The forests with the highest N deposition rate had mostly intermediately high organic layer N stocks, thus the N deposition rate and the N stock of the organic layer were only loosely related (Fig. 2c).

Tree stem growth was positively correlated with MAT ($R^2$=0.27), and it increased by a factor of 2.5 from 1.87 m$^3$ ha$^{-1}$ yr$^{-1}$ at -2.0°C MAT to 4.72 m$^3$ ha$^{-1}$ yr$^{-1}$ at 7.5°C (Fig. 2d).

The log-transformed C:N ratio of the organic layer was negatively correlated with MAT ($R^2$=0.17; Table 1) and with the log-transformed N deposition ($R^2$=0.21), and positively with latitude ($R^2$=0.16; Table1). In the linear model fitted to the data, the molar C:N ratio of the organic layer decreased from 55.3 at -2.0°C MAT to 37.8 at 7.5°C MAT (Fig. 2d).

The log-transformed P stock of the organic layer was positively correlated with MAT ($R^2$=0.13, Table 1) and with the log-transformed N deposition ($R^2$=0.12), and negatively with latitude ($R^2$=0.10; Table 1). In the linear model fitted to the data, the P stock of the organic layer increased by a factor of 2.3 from 28 kg ha$^{-1}$ at -2.0°C MAT to 65 kg ha$^{-1}$ at 7.5°C MAT (Fig. 2e).

The log-transformed N:P ratio of the organic layer was positively correlated with MAT ($R^2$=0.36, Table 1), and the log-transformed N deposition ($R^2$=0.38), and negatively with latitude ($R^2$=0.36). In the linear model fitted to the data, the N:P ratio of the organic layer increased by a factor of 2.1 from 24.0 at -2.0°C MAT to 49.3 at 7.5°C MAT (Fig. 2f).

The log-transformed C:P ratio of the organic layer was only weakly positively ($R^2$=0.05) correlated with MAT and the log-transformed N deposition, and negatively with latitude  (Table 1). Furthermore, C and N concentrations in the mineral soil were only weakly positively correlated with MAT and the log-transformed N deposition, and negatively with latitude (Table 1).

In most soils, the P concentration of the parent material was below 1.0 g kg$^{-1}$, but in a small number of forests, it was substantially higher, up to 3.4 g kg$^{-1}$. The P concentration of the soil parent material and the total P concentration of the upper 10 cm of the mineral soil were only very weakly negatively related with MAT and positively with latitude (Table 1).

## 3.2 Dominant tree species and SOM stoichiometry

The C:N ratio of the organic layer differed between forests with different dominant tree species (Fig. 3a). The C:N of the organic layer was highest in pine forests, on average 1.8 times higher than in deciduous forests, and 1.3 times higher than in spruce forests (Fig. 3a). A similar relationship between forests dominated by different tree species was observed for the C:N ratio in the mineral soil (Fig. 3b-d). In a depth of 55-65 cm, the C:N ratio of the mineral soil in spruce forests was on average 1.8 times higher than in deciduous forests and 1.2 times higher than in pine forests (Fig. 3d), similar to the organic layer. Some pine forests had very high C:N and C:P ratios due to low N and P contents. It needs to be taken into account that there is some uncertainty regarding the samples with very low N and P concentrations since the determination of N and P contents in very nutrient-poor samples is more strongly affected by sample inhomogeneity than the measurement of nutrient-rich samples.

The C:P ratio of the organic layer (Fig. 3e) followed a similar pattern as the C:N ratio across forests with different dominant tree species, and the C:P ratio of the organic layer in spruce forests was on average 1.3 times higher than in both deciduous and pine forests (Fig. 3e). In contrast, the C:OP ratio in the mineral soil was not elevated in pine forests but in deciduous forests (Fig 2f). The N:P ratio was elevated in deciduous forests, but did not significantly differ between pine and spruce forest (Supplement Fig. S2a). The organic layer stock differed also between forests with different dominant tree species (Supplement Fig. S2b). In deciduous forests, it was on average 2.2 times higher than in pine forests and 1.9 times higher than in spruce forests.

## 3.3 Texture and SOM stoichiometry

Concentrations of C, N and OP in the uppermost 10 cm of the mineral soil were highest in soils with the texture clay or fine silt (Fig. 4a-c). The concentrations of C, N and OP decreased gradually with increasing particle size (Fig. 4a-c). C, N and OP concentrations were higher in soils with the texture clay or fine silt than in soils with the texture sand or coarse sand by a factor of 2.3, 3.5, and 4.6, respectively (Fig 4a-c). Thus, the differences in element concentrations between soils with different texture were largest for OP, and decreased in the order OP>N>C. In contrast to OP, the total P concentration did not differ between soils with the texture class sand, find sand and coarse sand. Only soils with the texture clay had elevated P concentrations compared to soils of the other three texture classes by a factor of 1.9, but the differences between the texture classes were not statistically significant (p>0.05, Supplement Fig. S3). The C:N ratio and the C:OP ratio in the mineral soil (0-10 cm depth) increased gradually with increasing particle size (Fig. 4d and e). The mean C:N ratio and C:OP ratio were 1.5 and 1.4 times lower in soils with the texture clay or fine silt than in soils with the texture sand or coarse sand (Fig. 4d and e).

Different soil textures were also associated with differences in the N concentration of the organic layer (Fig. 4f). The mean N concentration of the organic layer increased with decreasing particle size, and it was 1.3 times higher in soils with the texture clay and fine silt than in soils with the texture sand and coarse sand (Fig. 4f).

### 3.4 Organic layer stock and element contents

The organic layer stock was inversely related with the concentration of exchangeable K in a non-linear way (Fig. 5a), and the variables were correlated when log-transformed ($R^2$=0.38; Fig. 5b). Further, the organic layer stock was also inversely related with its P concentration in a non-linear way (Fig. 5c), and the variables were correlated ($R^2$=0.27; Fig. 5d). The log-transformed organic layer stock was only weakly negatively correlated with the log-transformed P concentration of the parent material and

the P concentration of the uppermost 10 cm of the mineral soil (both $R^2$=0.03, $p<0.01$). The organic layer stock was also positively linearly related with the organic layer N:P ratio (Fig. 5e) and the log-transformed variables were correlated ($R^2$=0.25; Fig. 5f). In contrast, the organic layer stock was not significantly related to the N concentration of the organic layer, and only very weakly with its C:N ratio ($R^2$=0.03).

The log-transformed concentrations of exchangeable K and P of the organic layer were positively correlated ($R^2$=0.67,

Supplement Fig. S4a). In addition, the log-transformed concentrations of N and Mg in the organic layer were strongly positively correlated ($R^2$=0.81, Supplement Fig. S4b). In the organic layer, log-transformed C and N ($R^2$=0.87) as well as C and P ($R^2$=0.75) and N and P ($R^2$=0.83) stocks were strongly correlated (Supplement Fig. S5a, c, e). In the uppermost 10 cm of the mineral soil, log-transformed C and N concentrations were also strongly correlated ($R^2$=0.79, Supplement Fig. S5b) Further, the log-transformed C and P ($R^2$=0.47) and N and P ($R^2$=0.49) concentrations in the uppermost 10 cm of the mineral

were also positively correlated (Supplement Fig. S5d and f).

### 3.5 Covariance

Latitude and MAT were strongly negatively correlated ($R^2$=0.88). Further, the log-transformed N deposition rate was strongly negatively correlated with latitude ($R^2$=0.78), and strongly positively with MAT ($R^2$=0.82). Deciduous forests had a lower

latitude than all other forests, while mixed, pine, and spruce forests did not differ significantly in latitude (Fig. S6a). Further, the latitude of plots with different soil texture did not vary significantly (Fig. S6b). Spruce forests had a higher texture class, i.e., a finer soil texture than mixed and pine forests, while there was no significant difference in texture between deciduous, mixed, and pine forests (Fig. S6c).

### 3.6 Multiple regressions

We tested if we could improve the regressions between MAT and several soil properties (Table 1) by including data on texture, and the P concentration of the parent material (Table 2). The log-transformed C and total P concentrations of the mineral soil

(0-10 cm) could be better predicted by the combination of MAT and the log transformed P concentration of the parent material (Table 2) than by only MAT (Table 1). The prediction of the OP concentration of the mineral soil (0-10 cm) improved when

next to MAT and the log transformed P concentration of the parent material also soil texture was included (Table 2). Further, also the C:OP and N:OP ratios could be better predicted by the combination of MAT, soil texture, and the P concentration of the parent material (Table 2) than by only MAT (Table 1). For other variables, the improvement of the prediction was smaller, independent of whether the models included interactions between the independent variables or not.


## 4. Discussion

### 4.1 N stock increases strongly with increasing MAT

Our finding that the N stock of the organic layer increased more from -2.0°C to 7.5°C MAT than the organic layer stock (Fig.1a and b) indicates that the formation of the N stock is more sensitive to temperature than the formation of the organic matter

stock. The reasons for the strong change in the N stock with MAT could be, first, the higher N deposition towards the south/southwest of Sweden that enriches the organic layer in South/Southwest Sweden with N. Second, it could potentially be that the rate with which N is taken up by trees increases with decreasing MAT, leading to N depletion of the organic layer at sites with low MAT. Third, it might be that $N_2$ fixation decreases strongly with decreasing MAT. Fourth, it could be that the N:P ratio or the P concentration of the organic layer affects decomposition and net N mineralization. These four explanations

will be discussed in the following.

Atmospheric N deposition is well known to be an important N input in forest ecosystems in the south of Sweden (Akselsson et al., 2008, Högberg et al., 2017; Yu et al., 2018). However, our finding that N deposition was only loosely related to the N stock of the organic layer (Fig. 2c) suggests that N deposition only explains a small part of the increase in the N stock with increasing MAT. Yet, at the same time, we need to take into account that the N deposition data is relatively uncertain and

might not be fully representative of the accumulated differences in deposited N.

Concerning the second potential explanation, it needs to be considered that the growth rate of trees increased with MAT (Fig. 2d). Thus, the potential explanation that the organic layer N stock is decreased by a factor of 7.5 due to high tree N uptake in the forests with the lowest MAT and lowest tree growth rate compared to forests with the highest MAT and growth rate is rather questionable.

It seems most likely that the increase in the organic layer N stock with increasing MAT is related to an increase in $N_2$ fixation with increasing MAT. Rates of $N_2$ fixation can be as high as 4 kg N ha$^{-1}$yr$^{-1}$ in boreal ecosystems (DeLuca et al., 2002; Lagerström et al., 2007; DeLuca et al., 2008; Gundale et al., 2010), which is a larger N input than atmospheric N deposition in most boreal areas (Gundale et al., 2011) and in the forests studied here. $N_2$ fixation is known to depend strongly on temperature in temperate and boreal ecosystems (Sorensen et al., 2011; Gundale et al., 2012; Rousk et al., 2013). Therefore, it

seems likely that fact that the N stock of the organic layer increased more strongly with increasing MAT than the organic layer

stock is related to the temperature-dependence of $N_2$ fixation. Most studies that explored the temperature-dependence in high latitudinal ecosystems focused on arctic heath ecosystems (for exceptions see Gundale et al., 2012; Rousk et al., 2013), and explored the temperature-dependency of $N_2$ fixation in short-term incubation experiments, rather than natural gradients of MAT. Future research should fill this gap and explore the temperature-dependence of $N_2$ fixation in forest soils along MAT

gradients to investigate to which extent temperature-dependence of $N_2$ fixation explains the change in N stocks along the MAT gradient observed here.

Our finding that the N stock of the organic layer is positively related with MAT (Fig. 2a) suggests that N inputs to the soils increase more with MAT than the net N mineralization rate. Similarly, the increase in the organic layer stock with increasing MAT (Fig. 2b) suggests that MAT has a larger positive effect on plant productivity (Fig. 2d) than on decomposition.

Decomposition and N mineralization in the organic layer might be affected by the N:P ratio or the P concentration of the organic layer which both change considerably along the MAT gradient (Fig. 2e and f). Previous studies reported that C sequestration in the organic layer of forest soils was positively related with the organic layer N concentration (Berg and Matzner, 1997; Pregitzer et al., 2008; Janssens et al., 2010; Spohn, 2015). In the present study, we did not find a significant relationship between the organic layer stock and its N concentration. However, our study indicates that the N:P ratio or the P

concentration might affect organic matter decomposition and N mineralization, and hence both the organic layer stock and the N stock of the organic layer (as discussed further in 4.2).

Our results about the positive relationship between MAT and both the organic layer stock and the N stock of the organic layer are in accordance with Akselsson et al. (2005) showing stronger C accumulation in the organic layer in the South than in the North of Sweden and with Callesen et al. (2007) who studied 198 forest soils in North Europe and reported that the N stock of

the total soil profiles (organic layer plus uppermost 100 cm of the mineral soil) was positively correlated with MAT. Further, our finding that the C:N ratio of the organic layer decreases with increasing MAT (Supplement Fig. S1) is in accordance with a recent study on the organic layer C:N ratio in Swedish and German forests (Högberg et al., 2021).

**4.2 P and K contents are negatively related to the organic layer stock**

The relationship between the N:P ratio and the organic layer stock (Fig. 5e and f) is mostly caused by a negative relationship between the organic layer stock and its P concentration (Fig. 5c and d). The reasons for the negative relationship between the organic layer stock and its P concentration might be a higher contribution of fine woody debris, which is P-poor (Spohn et al., 2020b) to the organic layer in forests with high organic layer stock. Further, it could be that the decomposition rate of the organic layer is positively related with the organic layer P concentration. This is supported by a study showing that P is the

most limiting element for microbial activity during the first phase of decomposition of pine needles in Sweden (Staaf and Berg, 1982) and by a meta-analysis reporting that the decomposition rate decreased with increasing C:P ratio of plant detritus across different ecosystems (Zechmeister-Boltenstern et al., 2015).

Further, we found that the organic matter stock was negatively related with its concentration of exchangeable K (Fig. 5a and b). The reason for this negative correlation is likely that K leaches fast from plant litter (Osono and Takeda, 2004; Schlesinger et al., 2021), and thus, thin organic layers that consist largely of relatively young organic matter have a higher K concentration than massive organic layers that mostly consist of old, K-poor organic matter. In addition, it could be that there is a relatively high contribution of fine-woody debris, which is K-poor, to the more massive organic layers. The relationship found here between the K concentration of the organic layer and the organic matter stock (Figs. 4 and b) is in agreement with the relationship reported by Stendahl et al. (2017) between the C stock of the organic layer and both its K and Mn concentration.

### 4.3 Organic layer N:P ratio increases with increasing MAT

Our finding that the organic layer N:P ratio increased with increasing MAT (Fig. 2f) indicates that P is not incorporated into the organic layer in a constant ratio with N across all temperature regimes. The reason for the change in the N:P ratio with MAT could be a similar change in the N:P ratio of the plant litter inputs to the soils. This is supported by Akselsson et al. (2015) showing that the N:P ratio of pine and spruce needles was substantially larger in South/Southwest Sweden than in middle and North Sweden. This is further supported by studies about European forests reporting that the foliage N:P ratio increases with increasing N inputs, because trees do not take up enough P to balance the high N availability (Flückiger and Braun, 1998; Braun et al. 2010; Talkner et al. 2015, Jonard et al., 2015). Further, it was suggested in modelling exercises that high N inputs can lead to P limitation in south Swedish forests (Akselsson et al., 2008; Yu et al., 2018), resulting in increased needle N:P ratios (Yu et al., 2018). Changes in foliage N:P ratio with MAT likely translate into similar changes in the N:P ratio of the organic layer. The reason why P is not incorporated into the plant biomass in a constant ratio with N across all temperature regimes is likely that P availability limits P uptake at sites with high N availability. This is supported by studies showing that despite the young age of Swedish forest soils, a relatively large proportion of the soil P is adsorbed to Fe and Al oxides (Giesler et al., 1998; 2002; Tuyishime et al., 2022). In addition, also latitude-dependent differences in dominant tree species (i.e., more deciduous forests in the South) seem to contribute to the differences in the N:P ratio between different climate regimes since the N:P ratio of the organic layer was significantly higher in deciduous forests than in pine and spruce forests (Fig. S2a).

### 4.4 Organic matter stoichiometry varies with tree species, even in the subsoil

We found that the C:N and C:P ratios of the organic layer were higher in pine forests than in forests dominated by other tree species, and a very similar difference in the C:N ratio between forests dominated by different tree species was still observed in the mineral soil in a depth of 55-65 cm (Fig. 3). The differences in element ratios of the organic layer are likely derived from differences in the C:N and C:P ratio of the plant litter, since particularly the C:N ratios of the organic layer are closely associated with the element ratios of the plant litter in Swedish forest soils (Ladanai et al., 2010). Similarly in the mineral soil, the differences in the C:N ratio between forests dominated by different tree species are likely also due to the difference in the

C:N ratio of root litter and the litter layer (Cotrufo et al., 2013; Zechmeister-Boltenstern et al., 2015; Spohn and Chodak, 2015). In addition, the relatively low C:N ratio in the mineral soil in spruce forests compared to pine forests might also partially result from the fact that the spruce forests tended to have a slightly finer texture (Supplement Fig. S6c), which is associated with lower C:N ratios (Fig. 4d, see discussion below). This is in accordance with Stendahl et al. (2010) showing that in Swedish forests, spruce tends to grow in more fertile soils than pine. Furthermore, some of the pine forests had extremely high C:N and C:P ratios, which might be due to charcoal in the soils which is likely more abundant in pine forests than in other forests due to more frequent fires (Zackrisson, 1977). Previous studies also reported a very high C:N ratio in the organic layer and the uppermost cm of the mineral soil in pine forests compared to other forests in Europe (Vesterdal and Raulund-Rasmussen, 1998; Vesterdal et al., 2008; Hansson et al. 2011; Cools et al. 2014). However, this is the first study to show that this difference in the C:N ratio between forests dominated by different tree species is also visible in the mineral subsoil, in a depth of 55-65 cm, to our knowledge. Yet, it has to be considered that we cannot clearly attribute the differences in stoichiometry to differences in vegetation since pine forests might have been established preferably on soils that already had nutrient poor SOM. Further, we cannot exclude that differences in former land use (Goodale and Aber, 2001; Spohn et al., 2016) or the depth of the groundwater table also affect the mineral soil C:N ratio, although at least the former seems unlikely given that the mean stand age of the forests was 113 years.

The higher C:P ratio of the organic layer observed for pine forests than for all other forests is very likely also related to differences in the stoichiometry of the plant litter inputs. Our result that the C:P ratio of the organic layer was higher in pine forests than in all other forests is in accordance with similar observations of the difference in the organic layer C:P ratio between pine and spruce forests (Vesterdal and Raulund-Rasmussen, 1998; Ladanai et al., 2010). The reason why we found no substantial differences in the C:OP ratio in the mineral soil between forests dominated by different tree species (Fig. 3f) is that the concentration of OP in the mineral soil was strongly affected by texture (Fig. 4c), more strongly than the N concentration (Fig. 4b), as we will discuss in the following.

## 4.5 Organic P and N contents are high in fine-textured soils

The concentrations of C, N, and P in the uppermost 10 cm of mineral soils were higher in fine-textured soils than in coarse-textured soils. The reason for this is very likely the higher charge density of fine-textured than of coarse-textured soils that allows a large number of organic compounds to adsorb to charged mineral surfaces, which protects them against microbial decomposition through steric hindrance (Lützow et al., 2006, Kögel-Knabner et al., 2008, Kleber et al., 2015). Our results are in accordance with previous studies showing that the capacity of soils to store OM is largely determined by the proportion of fine mineral particles with high surface area and high charge density, such as for example, phyllosilicates and Fe and Al oxides (Oades, 1988; Mayer, 1994; Christensen, 1996; Hassink, 1997). Sorption of OM to mineral surfaces is one of the most important processes that slows down the decomposition of organic matter in soils (Lützow et al. 2006; Kleber et al. 2007; Kögel-Knabner et al. 2008), and very likely leads to enrichment of OM in the fine-textured forest soils studied here.

Our finding that the enrichment in fine-textured soils (compared to coarse-textured soils) increased in the order C<N<OP,
indicates that the capacity to compete for sorption sites is highest for P-containing organic compounds, and higher for N-containing compounds than for N- and P-free compounds. The reason for the strong enrichment of OP in fine-textured soils is likely that OP compounds adsorb rigidly to mineral surfaces. This interpretation is supported by studies demonstrating that phosphorylated organic compounds have a larger capacity to compete for binding sites in soils than non-phosphorylated organic compounds (Afif et al., 1995; Franssson and Jones, 2007; Schneider et al., 2010). Sorption to mineral surfaces makes OP compounds likely more persistent in soil than non-phosphorylated organic compounds (Spohn et al., 2020a, b). The fact that the total P concentration (Fig. S3) differed much less than the OP concentration (Fig. 4c) between soils of different texture classes supports our interpretation that the strong enrichment of OP in the fine-textured soils is mostly caused by rigid adsorption of OP compounds (that protects OP against decomposition) and much less by a higher P concentration or P availability in these soils. Similarly, a large part of N is present in SOM in the form of charged nitrate or ammonium moieties, for example in peptides that can adsorb to charged surfaces, which decreases their decomposition rate (Knicker et al., 1993; Jones and Hodges, 1999; Miltner et al., 2009; Kopittke et al., 2018). Thus, the large capacity of N- and P-containing organic compounds to adsorb to charged surfaces in soils is likely the reason why N, and particularly OP has higher concentrations in the fine-textured soils compared to the coarse-textured soils than C. In addition, regarding the N concentration it could also be that fine-textured soils are commonly formed from nutrient-rich (potassium, phosphorus, magnesium, etc.) minerals which causes high plant productivity and $N_2$ fixation, resulting in higher N concentrations in fine-textured soils compared to coarse-textured soils (Clarholm and Skyllberg, 2013). This is likely also the reason for the higher N concentration of the organic layer in the fine-textured soils compared to the coarse-textured soils (Fig. 4f).

## 4.6 Future research questions

Based on the results gained from the analysis of forest soil inventory data, we identified the following questions that should be studied in the future.

First, our finding that the N:P ratio of the organic layer increased strongly with increasing MAT and the atmospheric N deposition rate raises the question if growth of trees in Scandinavia at sites with high atmospheric N deposition is limited by P, and if so, to what extent.

Second, future research should study the temperature-dependence of $N_2$ fixation in forest soils in Scandinavia along MAT gradients to investigate to which extent temperature-dependence of $N_2$ fixation explains the change in N stocks along the MAT gradient observed here.

Third, the result that the OP concentration in the mineral soil depends strongly on soil texture, which is likely due to rigid adsorption of OP compounds on soil minerals, calls for future investigations of (a) the role of OP for the sorptive stabilization of SOM and (b) the turnover of the soil OP pool in relation to the soil organic C pool.

Fourth, we speculated that very high C:N ratios in some pine forests might be related to forest fires. Future research should explore the legacy of forest fires on SOM stoichiometry.

Fifth, the negative relationships found here between the organic layer stock and the organic layer P concentration raises the question if and to which extent P limits organic matter decomposition in Scandinavian forests.


## 4.7 Conclusion

We found that the N stock of the organic layer increased more with MAT than the organic layer stock, which is might be due to the temperature-dependency of $N_2$ fixation and the P-dependency of decomposition. Against our first hypothesis, we observed that the N:P ratio of the organic layer increased substantially with increasing MAT, likely due to an increase
availability of N relative to P with increasing MAT. Further, the C:P ratio showed similar differences between forests dominated by different tree species as the C:N ratio in the organic layer, as hypothesized, but the C:OP ratio in the mineral soil differed little between forests dominated by different tree species. The reason for this is likely that the OP concentration was very strongly affected by texture in the mineral soil, as the concentration of OP was much higher in fine- than in coarse-textured soils. The difference in element concentrations between fine- and coarse-textured soils decreased in the order
OP>N>C, in agreement with the third hypothesis. Taken together, the results show that the N:P ratio of the organic layer was most strongly related to MAT. Further, the C:N ratio was most strongly related to dominant tree species, even in the mineral subsoil. In contrast, the C:P ratio was only affected by dominant tree species in the organic layer, while the C:OP ratio in the mineral soil was hardly affected by tree species due to the strong effect of soil texture on the OP concentration.

**Code Availability**

The R code will be made publically available once the manuscript is accepted for publication, and is available for editor and reviewers as an asset to this manuscript.

**Data availability**

All data will be made publically available once the manuscript is accepted for publication, and is available for editor and reviewers as an asset to this manuscript.

**Author contributions**

MS designed the study, conducted the data analysis, and wrote the manuscript, JS leads the Swedish Forest Soil Inventory and
contributed to the manuscript.

**Acknowledgments**

The authors thank all technical staff who conducted the sampling and chemical analyses, and particularly Oscar Skirfors for conducting the P analyses. The Swedish Forest Soil Inventory is part of the national environmental monitoring commissioned
by the Swedish Environmental Protection Agency. The authors thank three anonymous reviewers for their helpful comments.

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

## Tables

**Table 1:** Results of the regression analyses (adjusted $R^2$, p value) for the organic layer, mineral soil (0-10 cm), and parent material in 309 Swedish forest soils (n=309). The signs + and - indicate positive and negative regressions, respectively.

| | Latitude (°) | MAT (°C) | MAP (mm) | Log (N deposition) (kg ha$^{-1}$) |
|---|---|---|---|---|
| Log (Organic layer stock) (t ha$^{-1}$) | - 0.20, p<0.001 | + 0.24, p<0.001 | + 0.02, p<0.01 | + 0.23, p<0.001 |
| Log (C stock organic layer) (kg ha$^{-1}$) | - 0.16, p<0.001 | + 0.18, p<0.001 | + 0.02, p<0.05 | + 0.17, p<0.001 |
| Log (N stock organic layer) (kg ha$^{-1}$) | - 0.26, p<0.001 | + 0.30, p<0.001 | + 0.04, p<0.001 | + 0.30, p<0.001 |
| Log (P stock organic layer (kg ha$^{-1}$) | - 0.10, p<0.001 | + 0.13, p<0.001 | + 0.01, p<0.05 | + 0.12, p<0.001 |
| Log (Molar C:N ratio organic layer) | + 0.16, p<0.001 | - 0.17, p<0.001 | - 0.04, p<0.001 | - 0.21, p<0.001 |
| Log (Molar C:P ratio organic layer) | - 0.06, p<0.01 | + 0.05, p<0.01 | p>0.05 | + 0.05, p<0.01 |
| Log (Molar N:P ratio organic layer) | - 0.36, p<0.001 | + 0.36, p<0.001 | + 0.06, p<0.001 | + 0.38, p<0.001 |
| Log (C concentration mineral soil) (g kg$^{-1}$) | - 0.17, p<0.001 | + 0.18, p<0.001 | p>0.05 | + 0.14, p<0.001 |
| Log (N concentration mineral soil) (g kg$^{-1}$) | - 0.08, p<0.001 | + 0.10, p<0.001 | p>0.05 | + 0.10, p<0.001 |
| Log (OP concentration mineral soil) (g kg$^{-1}$) | p>0.05 | p>0.05 | p>0.05 | p>0.05 |
| Log (Molar C:N ratio mineral soil) | p>0.05 | p>0.05 | p>0.05 | p>0.05 |
| Log (Molar C:OP ratio mineral soil) | - 0.11, p<0.001 | + 0.12, p<0.001 | p>0.05 | + 0.10, p<0.001 |
| Log (Molar N:OP ratio mineral soil) | - 0.15, p<0.001 | + 0.15, p<0.001 | p>0.05 | + 0.13, p<0.001 |
| Log (P concentration mineral soil) (mg kg$^{-1}$) | + 0.03, p<0.01 | - 0.01, p<0.05 | - 0.02, p<0.01 | - 0.00, p<0.05 |
| Log (P conc. parent material) (mg kg$^{-1}$) | + 0.09, p<0.001 | - 0.06, p<0.001 | - 0.01, p<0.05 | - 0.05, p<0.001 |

**Table 2:** Results of the multiple regression analyses (adjusted $R^2$, p value) with (*) and without (+) interactions of the independent variables for the organic layer and mineral soil (0-10 cm) in 309 Swedish forest soils (n=309).

| | MAT + Texture | MAT * Texture | MAT + Log (Pparent) | MAT * Log( Pparent) | MAT + Texture + Log (Pparent) | MAT * Texture * Log (Pparent) |
|---|---|---|---|---|---|---|
| Log (Organic layer stock) (t ha$^{-1}$) | 0.24, p<0.001 | 0.24, p<0.001 | 0.24, p<0.001 | 0.24, p<0.001 | 0.24, p<0.001 | 0.24, p<0.001 |
| Log (C stock organic layer) (kg ha$^{-1}$) | 0.18, p<0.001 | 0.18, p<0.001 | 0.19, p<0.001 | 0.18, p<0.001 | 0.18, p<0.001 | 0.18, p<0.001 |
| Log (N stock organic layer) (kg ha$^{-1}$) | 0.30, p<0.001 | 0.30, p<0.001 | 0.30, p<0.001 | 0.30, p<0.001 | 0.30, p<0.001 | 0.30, p<0.001 |
| Log (P stock organic layer (kg ha$^{-1}$) | 0.14, p<0.001 | 0.14, p<0.001 | 0.13, p<0.001 | 0.13, p<0.001 | 0.14, p<0.001 | 0.14, p<0.001 |
| Log (Molar C:N ratio organic layer) | 0.21, p<0.001 | 0.21, p<0.001 | 0.19, p<0.001 | 0.21, p<0.001 | 0.21, p<0.001 | 0.23, p<0.001 |
| Log (Molar C:P ratio organic layer) | 0.07, p<0.001 | 0.08, p<0.001 | 0.09, p<0.001 | 0.09, p<0.001 | 0.08, p<0.001 | 0.09, p<0.001 |
| Log (Molar N:P ratio organic layer) | 0.35, p<0.001 | 0.35, p<0.001 | 0.36, p<0.001 | 0.37, p<0.001 | 0.36, p<0.001 | 0.37, p<0.001 |
| Log (C concentration mineral soil) (g kg$^{-1}$) | 0.23, p<0.001 | 0.23, p<0.001 | 0.22, p<0.001 | 0.27, p<0.001 | 0.27, p<0.001 | 0.26, p<0.001 |
| Log (N concentration mineral soil) (g kg$^{-1}$) | 0.26, p<0.001 | 0.26, p<0.001 | 0.20, p<0.001 | 0.20, p<0.001 | 0.30, p<0.001 | 0.28, p<0.001 |
| Log (OP concentration mineral soil) (g kg$^{-1}$) | 0.11, p<0.001 | 0.12, p<0.001 | 0.08, p<0.001 | 0.08, p<0.001 | 0.18, p<0.001 | 0.18, p<0.001 |
| Log (Molar C:N ratio mineral soil) | 0.05, p<0.001 | 0.05, p<0.001 | >0.05 | >0.05 | 0.05, p<0.001 | 0.06, p<0.01 |
| Log (Molar C:OP ratio mineral soil) | 0.15, p<0.001 | 0.16, p<0.001 | 0.13, p<0.001 | 0.13, p<0.001 | 0.16, p<0.001 | 0.17, p<0.001 |
| Log (Molar N:OP ratio mineral soil) | 0.15, p<0.001 | 0.16, p<0.001 | 0.17, p<0.001 | 0.17, p<0.001 | 0.16, p<0.001 | 0.18, p<0.001 |
| Log (P concentration mineral soil) (mg kg$^{-1}$) | 0.03, p<0.01 | 0.03, p<0.01 | 0.20, p<0.01 | 0.20, p<0.01 | 0.19, p<0.01 | 0.20, p<0.01 |



# Figures

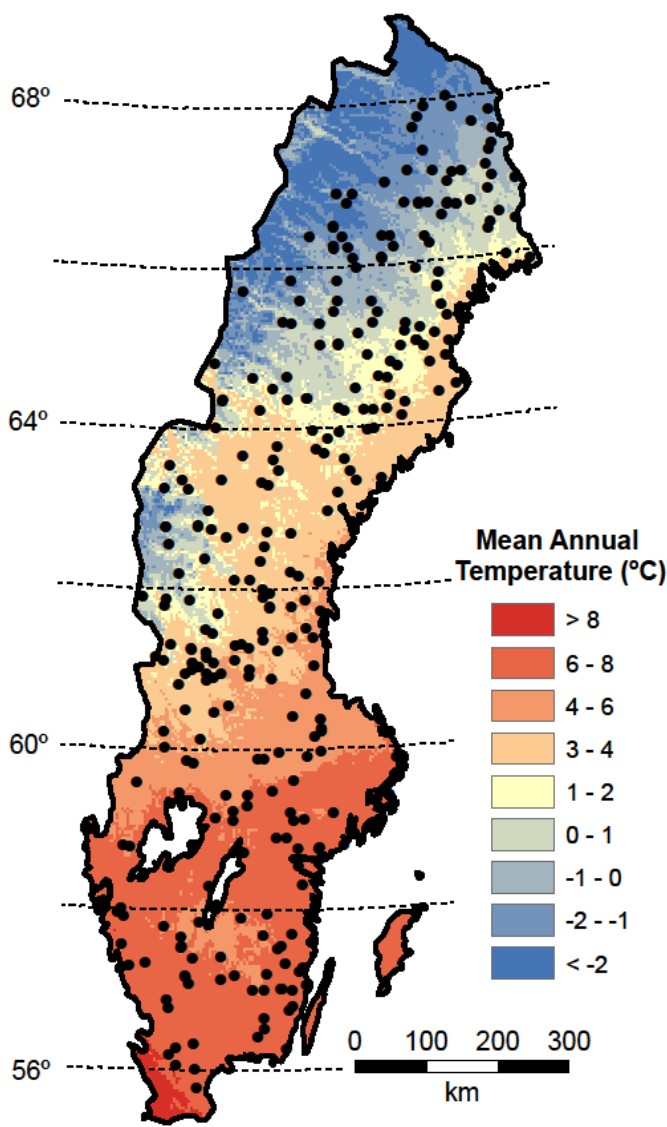

**Figure 1:** Map depicting mean annual temperature (MAT), latitude, and the plot locations (black dots).

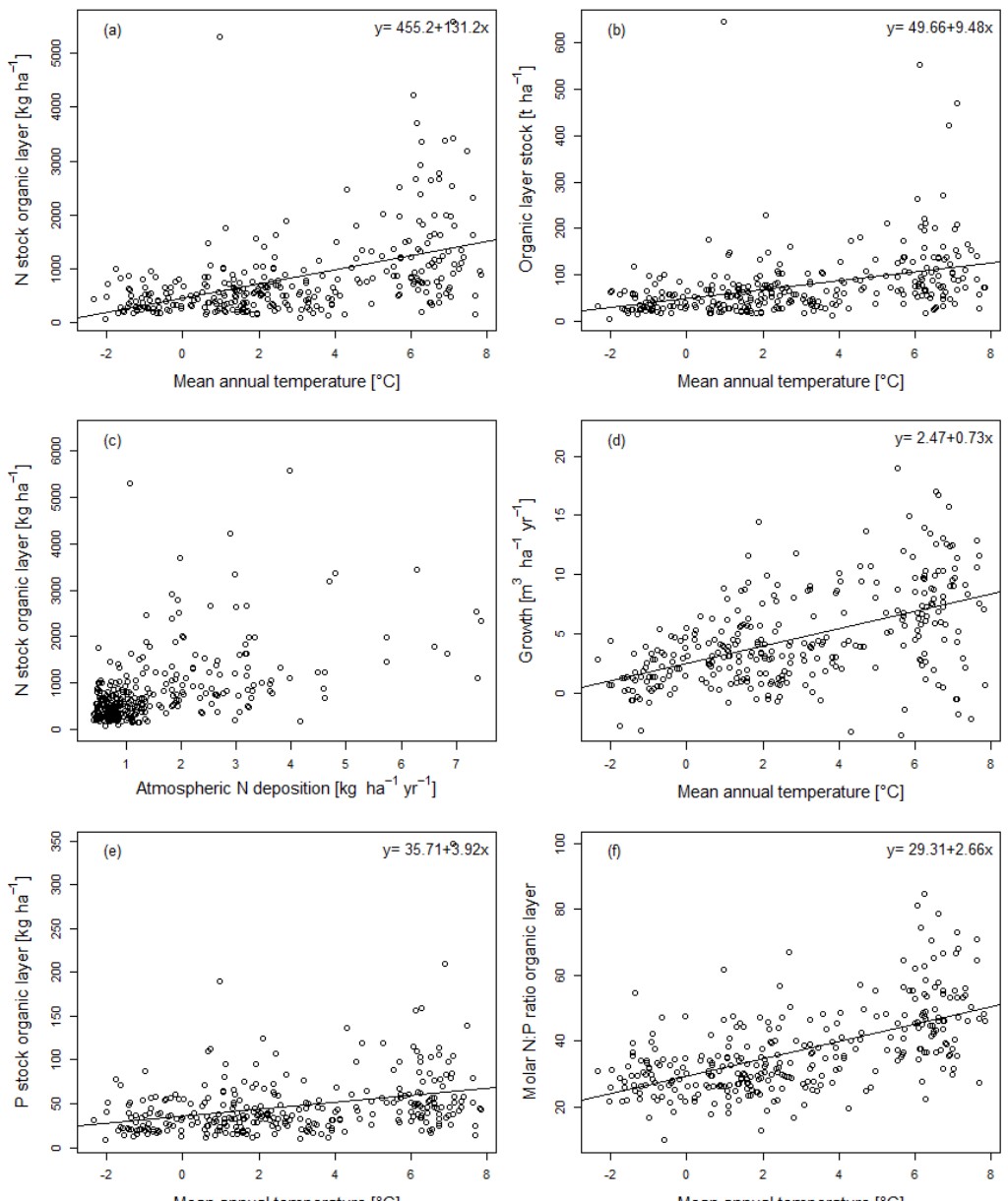

**Figure 2:** The nitrogen (N) stock of the organic layer (a) and the organic layer stock (b) as a function of mean annual temperature together with the N stock of the organic layer as a function of atmospheric N deposition (c) as well as tree stem growth (d), the P stock of the organic layer (e), and the molar nitrogen-to-phosphorus (N:P) ratio of the organic layer (f) as a function of mean annual temperature in 309 Swedish forest soils with a stand age >60 years.

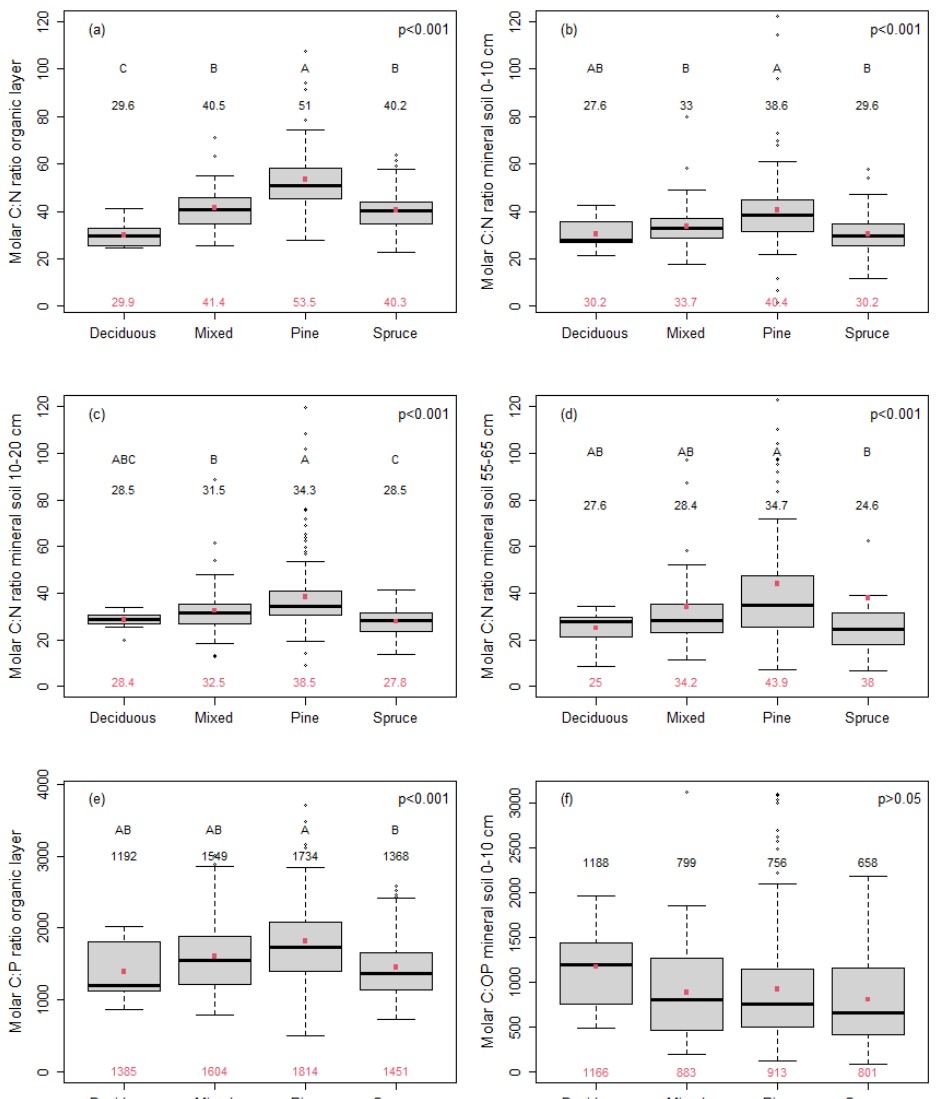

**Figure 3:** Molar carbon-to-nitrogen (C:N) ratio of the organic layer (a) and the mineral soil in 0-10 cm depth (b), 10-20 cm depth (c), and 55-65 cm depth (d) as well as the molar carbon-to-phosphorus (C:P) ratio of the organic layer (e) and the molar carbon-to-organic phosphorus (C:OP) ratio in the mineral soil in 0-10 cm (f) depending on the dominant tree species (deciduous n=10, mixed n=67, pine n=144, and spruce n=88). Please notice that mixed refers to mixed pine and spruce forest. Black numbers give the median, red dots and red numbers depict the arithmetic mean. Different capital letters indicate statistically significant differences ($p < 0.05$) between plots with different dominant tree species, while the p value of the ANOVA is indicated in the right corner of each panel.

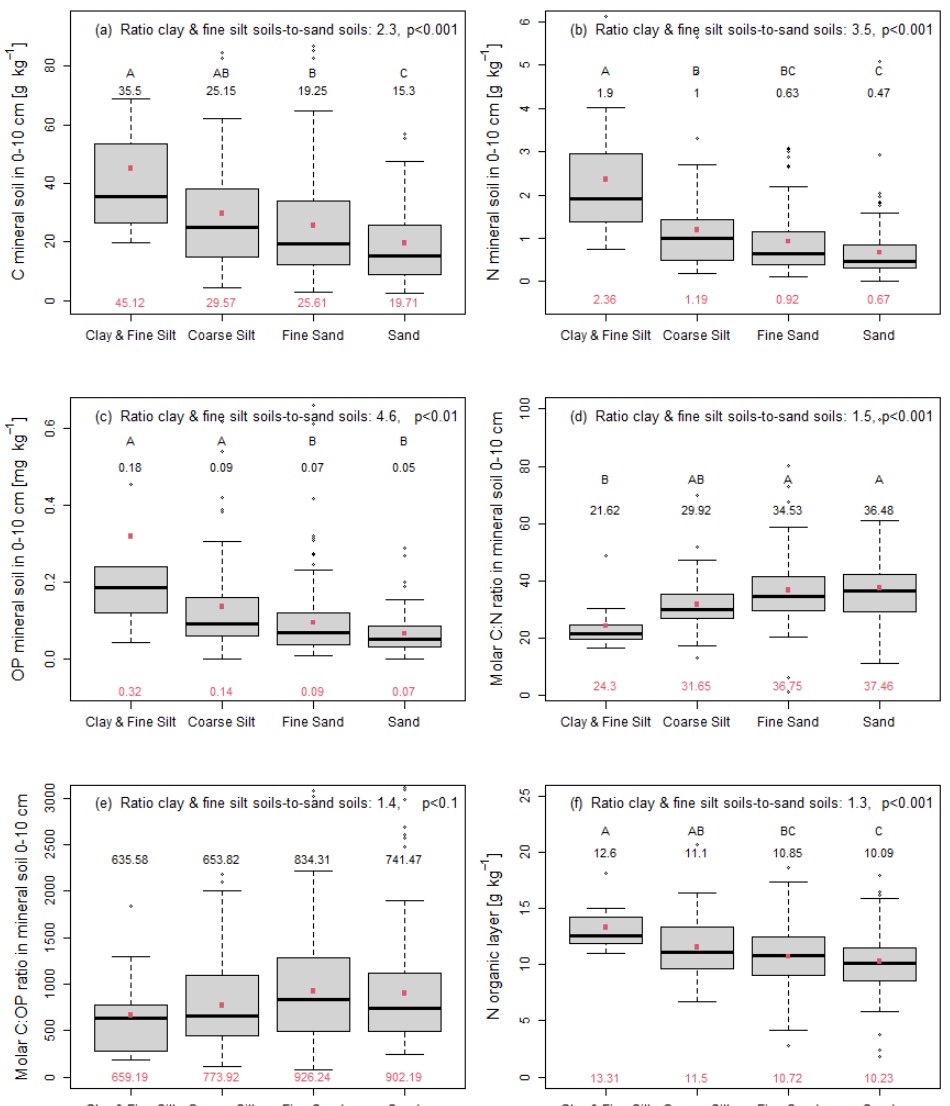

**Figure 4:** Carbon (a), nitrogen (b) and organic phosphorus (c) concentration, the molar carbon-to-nitrogen (C:N) ratio (d), and the molar carbon-to-organic phosphorus (C:OP) ratio (e) in the mineral soil in a depth of 0-10 cm as well as the nitrogen (N) concentration of the organic layer (f) depending on the soil texture (clay and fine silt n=11, coarse silt n=52, fine sand n=136, and sand n=110) in 309 Swedish forest soils with a stand age >60 years. The texture class called sand encompasses sand and coarse sand. The indicated ratio of clay and fine silt soils-to-sand soils is the mean ratio. Black numbers give the median, red dots and red numbers depict the arithmetic mean. Different capital letters indicate statistically significant differences (p<0.05) between the soils of different texture classes, while the p value of the ANOVA is indicated in the right corner of each panel.



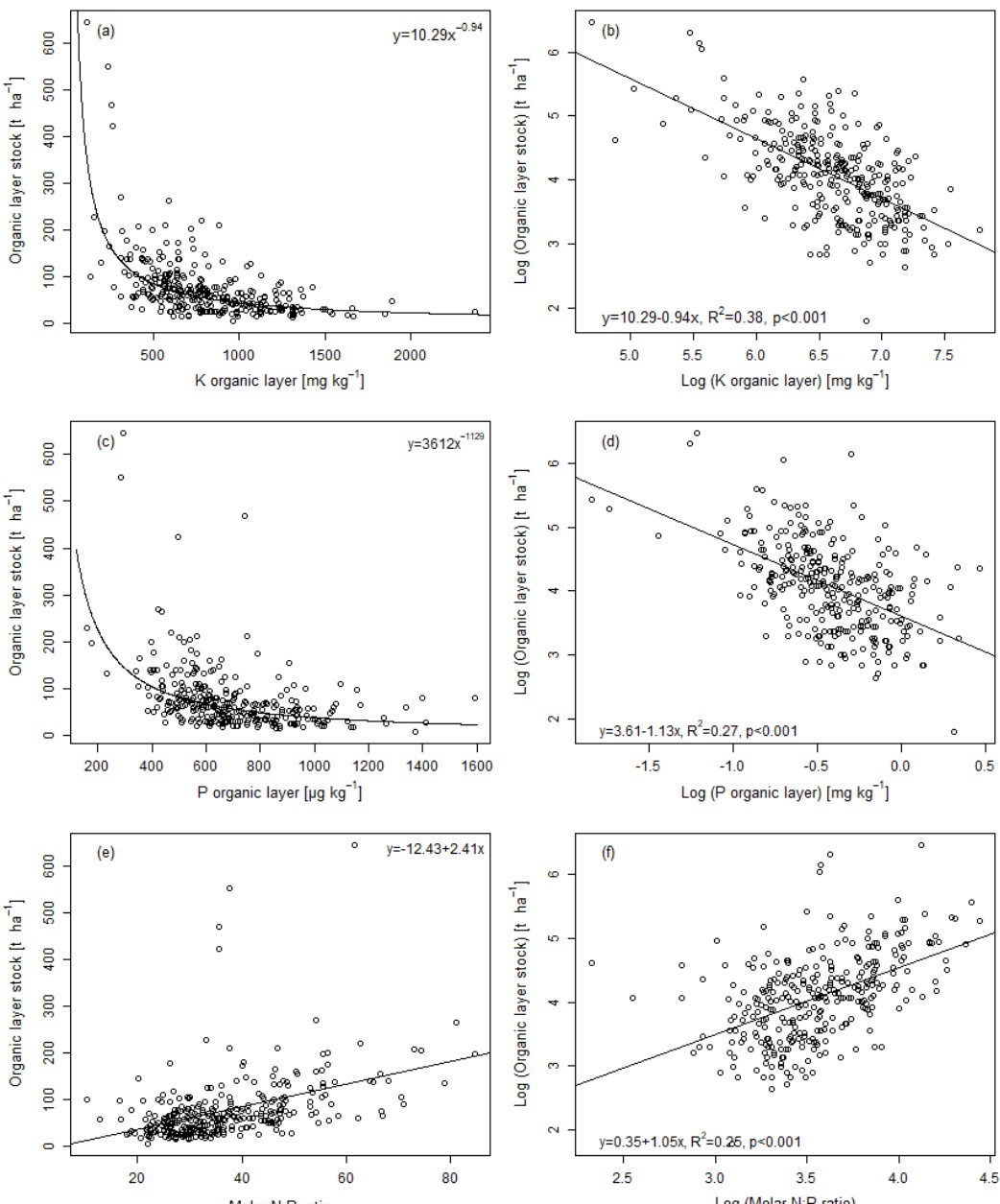

**Figure 5:** The organic layer stock as a function of the organic layer K concentration shown for the original data with a power function (a) and the log-transformed data with a linear model (b) as well as the organic layer stock as a function of the organic layer P concentration shown for the original data with a power function (c) and the log-transformed data with a linear model (d), and the organic layer stock as a function of the organic layer molar N:P ratio shown for the original data (e) and the log-transformed data with a linear model (f).