# Peer review of "Carbon, nitrogen, and phosphorus stoichiometry of organic matter in Swedish forest soils and its relationship with climate, tree species, and soil texture"

_Biogeosciences, 2021_

## Referee Comment (RC3)

Review LTS

Title: Carbon, nitrogen, and phosphorus stoichiometry of organic matter in Swedish forest soils and its relationship with climate, tree species, and soil texture
Author(s): Marie Spohn and Johan Stendahl MS No.: bg-2021-346 MS type: Research article

General comments:
There is an unredeemed potential for interesting publications in the large databases of national forest/soil inventories. This is also the case for the Swedish Forest Soil Inventory, publications using these data are highly welcome. The present study focuses on carbon (C), nitrogen (N)and phosphorus (P) and the ratios C:N, C:P, and N:P in both organic and mineral soil layers which particularly the P and the ratio aspect are novel. After reading the paper, I am however, a bit puzzled. The manuscript/tittle and objective promises to explore relationship with climate, tree species and soil texture, and it delivers on showing these relationships, but the discussion offers very little or patchy interpretation. My greatest concern with regard to this manuscript is the lack of discussion of the results and relationship against important factors known to regulate the accumulation/loss of C from soil -the balance of input of organic material against decomposition. Instead the main part of the discussion is linked to changes in N and here the focus is on three hypotheses *" first, the higher N deposition towards the south/southwest of Sweden that enriches the organic layer in South/Southwest Sweden with N. Second, it could potentially be that the rate with which N is taken up by trees increases with decreasing MAT, leading to N depletion of the organic layer at sites with low MAT. Third, it might be that N2 fixation decreases strongly with decreasing MAT."* While I do not dispute that these could be relevant hypotheses, I find it strange that forest productivity and decomposition processes are not discussed more in full. The same criticism could be directed towards the discussion of the P, here I had expected that the regional distribution of P in parent material (geology) would have included, also a reflection regarding limits P nett-immobilisation and nett-mineralisation. The text is also drawn down by some unnecessary errors which could easily have been corrected before submission (see comments below). My conclusion is therefore that the manuscript is not ready for publishing, there are too many errors, and insufficient discussion of the results, see comments below for more specific reasons for my conclusion.

1.      Does the paper address relevant scientific questions within the scope of BG?
        a.  yes
2.      Does the paper present novel concepts, ideas, tools, or data?
        a.  Novel concepts: to some degree
        b.  Novel Ideas: the background for the first hypotheses is not clearly stated, but with better explanation it could be considered novel. I suggest that the authors should consider e.g. Sundqvist, M. K., Liu, Z. F., Giesler, R., Wardle, D. A., 2014, Plant and microbial responses to nitrogen and phosphorus addition across an elevational gradient in subarctic tundra, Ecology 95(7):1819-1835. Which presents N:P ratios in relationship with temperature gradient that coincide with the findings in the current study.
        c.  Novel tools: no
        d.  Novel data: yes
3.      Are substantial conclusions reached?
        a.  To some degree (hypothesis 1 is not treated satisfactory)
4.      Are the scientific methods and assumptions valid and clearly outlined?
        a.  More or less - with some concerns the most important listed here:

b. How are the ratios calculated, are they based on mass (concentration) or are they base on Molar ratio. The standard method in soil science is on mass - this is not that apparent for C:N ratio but will have large implications when calculating C:P and N:P ratios. Clarity in method is essential to know when comparing with other studies - though the relationship with MAT, tree species, texture will not be affected. Please check the C:P ratios given in figure 3e.

c. What is "organic lay stock" and how is it calculated

d. Several analyses that are referred to in the materials and methods are not used in the texts, Two of them I would hope you have considered and probably you have reason for not including - the pH and the P in the parent material?

5. Are the results sufficient to support the interpretations and conclusions?
   a. Yes and No, see specific comments below

6. Is the description of experiments and calculations sufficiently complete and precise to allow their reproduction by fellow scientists (traceability of results)?
   a. Some improvements needed - se comments below

7. Do the authors give proper credit to related work and clearly indicate their own new/original contribution?
   a. yes

8. Does the title clearly reflect the contents of the paper?
   a. yes

9. Does the abstract provide a concise and complete summary?
   a. The abstract only refers to relative distributions - I would recommend to include the range or levels of the ratios, concentrations, stocks.

10. Is the overall presentation well-structured and clear?
    a. Yes, in general

11. Is the language fluent and precise?
    a. Yes in general, see comments below

12. Are mathematical formulae, symbols, abbreviations, and units correctly defined and used?
    a. Formulas for calculation of stocks is missing and should be better explained, Also a specification of how the ratios were calculated should be included - from the figures I gather it is Molar ratio, but in the text it is not specified (both ratio and molar ratio is referred to)

13. Should any parts of the paper (text, formulae, figures, tables) be clarified, reduced, combined, or eliminated?
    a. See comments below

14. Are the number and quality of references appropriate?
    a. Yes, sufficient

15. Is the amount and quality of supplementary material appropriate?
    a. yes

Specific comments:

L 15-16, repetition of >60 years- consider deleting one place or merging the two sentences

L17, and L26 "organic layer stock" stock of what? should be it be "organic layer C stock"? or are we talking about layer thickness? volume? density? Mass?

Line 19: "went along with" - better "followed"

L24-25 "OP adsorbs very rigidly to mineral surfaces." Could this not also be an effect of higher P availability in fine textured soils?

Line26. "C and N concentration in the mineral soil" is a rather unprecise term as these concentrations normally are measured in several different horizons and are expected to differ

significantly with depth and genetic horizon. If you were talking about stocks however, the correlations would appear more valid/informative.

L51: «most soils in Northern Europe are only between 9,000 and 14,000 years old". I believe you should then define Northern Europe - you need only to go to Denmark to find older soils. Perhaps I it is better to write "Sweden" or "Scandinavia" instead of "Northern Europe".

L50-55. It would be interesting to know at what limit, N:P ratio, you would expect P to become limiting.

L78-80 "The soils are a representative subsample of Swedish forest soils with a stand age >60 years that were sampled for the Swedish Forest Soil Inventory. We selected sites with a stand age >60 years in order to minimize the effect of forest management on the results" will these cover both self-drained and poorly drained soils?

L104 - what are the numbers in the brackets referring to? - are they necessary?

L 105-106: "and living and dead roots >1 mm diameter are removed." This is nearly an impossible task on dried soils - and according to personal experience would also introduce a bias - thin roots are easier to remove from coarse textured soils than from fine textured soils. I suggest that a better description would be " (where possible) living and dead roots were removed from the sample prior to sieving (<2mm) and homogenizing" of cause only if this is the case.

L109 -use "covered" instead of "vegetated"

L113-114: "each three soils were classified as either leptosol, gleysol or cambisol, while one soil was an umbrisol" - Here I suggest a rephrasing: 3 were leptosol, 3 were gleysol, 3 were cambisol, and one was umbrisol

L145: unclear "Organic layer element stocks were calculated by multiplying the organic layer stock with the respective element concentration." What is the "organic layer stock"? how is it calculated?

L168-169: There must be an error "The C:N ratio of the organic layer was negatively correlated with MAT ($R_2=0.08$, Supplement Fig. S1) and with the log-transformed N deposition ($R_2=0.11$) and latitude ($R_2=0.08$; Table1)." C:N ratios cannot be negatively correlated to latitude - they must be positive. It would have been better to have r values in table 1 showing either negative or positive correlations not just "$R^2$".

L169-170: The molar C:N ratio is referred to for the first time -- are all ratios molar ratios? - not described in the M&M

L178-179: "Furthermore, C, N and OP in the mineral soil were only weakly correlated with latitude, MAT, and N deposition (Table 1)." Table may show correlations but no indication whether positive or negative.

L185." the C:N ratio of the mineral soil in spruce forests" should it not be pine? The same error applies for L189 with regard to C:P ratio

L187: "Some pine forests had very high C:N and C:P ratios due to low N and P contents." In the materials and methods you should introduce some rules for discarding "unusual" ratios based on the detection limits of the elements. These ratios are "artifact" and any interpretation of these may be useless.

L192-194, "The organic layer stock differed also between forests with different dominant tree species (Supplement Fig. S2b). In deciduous forests, it was on average 2.2 times higher than in pine forests and 1.9 times higher than in spruce forests." This does not sound right - what is "organic layer stock"?

L202-204 : "The mean C:N ratio and C:OP ratio were 1.5 and 1.4 times higher in soils with the texture clay or fine silt than in soils with the texture sand or coarse sand (Fig. 4d and e)." Error - the opposite → The mean C:N ratio and C:OP ratio were 1.5 and 1.4 times **lower** in soils with the texture clay or fine silt than in soils with the texture sand or coarse sand (Fig. 4d and e).

L244-246. Both "organic layer stock" and "organic matter stock" are used about the same parameter - stick to one (I believe it should be either better described and defined as well, see earlier comments)

L253 254: "the stem growth rate of trees **decreased** with MAT (Fig. 2d)" but figure 2 d show that the growth rate **increases** with MAT!

L279 "foliage N:P ratio **decreases** with increasing N inputs" should be increase?

L307 "miner" should be "mineral"?
L309-310 " However, this is the first study to show that this difference in the C:N ratio between forest dominated by different tree species is also visible in the mineral subsoil, in a depth of 55-65 cm, to our knowledge." But is this really confirmed by figure 3d? How have you handled the very low concentrations and detection limits?

L320 "changed" should be "charged",
L337 "changed" should be "charged"

L320-340 -4.4. Organic P and N contents are high in fine-textured soils, I find this paragraph too superficially discussed.

L337: "Thus, the high capacity of N- and P-containing organic compounds to adsorb to changed surfaces in soils is likely the reason why N, and particularly OP has higher concentrations in the fine-textured soils compared to the coarse-textured soils than C." - this is not only due to sorption. These fine textured soils are more productive nutrient rich system - (P -rich included) - also the water holding capacity and pH increases fine textured soils. and what about regional distribution of P containing minerals?

L345 - 355 : 4.5 P and K contents are negatively related to the organic layer stock. The discussion in this paragraph is poor. These two elements behave very different in soil the potassium is a cation more or less only found in inorganic forms, while P mostly would be present as a anionic molecule and in organic form be part of organic molecule . K+ could easily be leached, even more so in acid conditions, while P-is retained and we would not expect any "fast leaching of P" I would find it highly unlikely that "P follows the same dynamic as K"

L 365 "the results show that the N:P ratio of the organic layer was most strongly related to MAT" showing a positive relationship, while the CN ratio showed a negative relationship with MAT. This confirmes that the boreal forest are N deprived, any increase in N availability will increase forest productivity P is not the limiting factor

L570 figure 3e  the molar C:P ratio of the organic layer is high - are you sure they are correct?

L570 figures b, c, d, and f, why do some of the means depict "Inf"?

---

## Author Comment (AC1)

**Answers to reviewer#1**

GENERAL COMMENTS

Spohn and Stendahl collected organic matter P data to complement a dataset of the Swedish Forest Soil Inventory. Based on these updated SFSI data, the authors investigated links between organic layer + mineral soil SOM C:N:P stoichiometry, vs climate, tree species and soil texture. Main findings of the study were that SOM C:N ratio differed among the dominant tree species, more so than was the case for the C:OP ratio. Especially in the deeper soil, the relationship between C:OP and species was weaker than for C:N, because of stronger associations of OP than N with finer soil particles (i.e. influence of texture, rather than assumed plant-soil feedbacks). N:P ratio increased with MAT, with potential implications for relative element availabilities. The study can be classified under the field of biogeochemistry, and assumes plant-soil feedbacks, which perfectly match the scope of *Biogeosciences*.

The manuscript is concise, very well structured, and easily readable with its subsections. I really appreciate the attention for P in a region where the nutrient is usually neglected because N limitation is assumed. Studying P, even in Nordic forests, is relevant because P can influence ecosystem function through interactions with other biogeochemical cycles, and P can be (co-)limiting in valleys and some sub-regions. *We thank the reviewer for positive evaluation of the manuscript and the detailed comments.*

I have the following major comments to be addressed, further explained in the 'specific comments' section:

- The manuscript is concise and the authors remain close to the data in the Discussion. This avoids too much speculation, but I do strongly recommend writing a short 1-2 paragraph section 4.6 with implications the authors see for future research. For example: what is the relevance of the findings in the context of nutrient availability/limitation research? See also the specific comment on *Line 356*, ...

*We added a section called 4.6 Future research questions in the end of the Discussion that reads as follows:*

*"4.6 Future research questions*
*Based on the results gained from the analysis of the forest soil inventory, we identified the following questions that should be studied in the future.*
*First, our finding that the N:P ratio of the organic layer increased strongly with increasing MAT and the atmospheric N deposition rate raises the question if growth of trees in Scandinavia at sites with high atmospheric N deposition is limited by P, and if so, to what extent.*
*Second, future research should study the temperature-dependence of N2 fixation in forest soils in Scandinavia along MAT gradients to investigate to which extent temperature-dependence of N2 fixation explains the change in N stocks along the MAT gradient observed here*
*Third, the result that the OP concentration in the mineral soil depends strongly on soil texture, which is likely due rigid adsorption of OP compounds on soil minerals calls for future investigations of (a) the role of OP for the sorptive stabilization of SOM and (b) the turnover of the soil OP pool in relation to the soil organic C pool.*
*Fourth, we speculated that very high C:N ratios in some pine forests might be related to forest fires. Future research should explore the legacy of forest fires on SOM stoichiometry.*
*Fifth, the negative relationships found between organic layer stock and organic layer P concentration raises the question if and to which extent P limits organic matter decomposition in Scandinavian forests.*
*"*

- Some choices on data selections, classifications and statistics are not well motivated or explained. For example, were mull-humus soils just absent, or excluded? Why? Why were mixed forests categorized with spruce for the multiple regressions? Why were some positively-skewed variables log-transformed for the analyses and others not? Answers to these questions potentially (co-)explain some of the patterns I can see in the figures. See specific comments for more detail and further examples.

*We added some more explanations throughout the Material and Methods section:*

*We excluded plots with the humus form mull because at these plots, the organic layer is not sampled separately from the mineral soil in the SFSI. We now added this information in the Material and Method section.*

*In the reviewed version of the manuscript, we had tried to include the variable dominant tree species into the multiple regression analysis, by assigning numbers to the trees species. However, we now removed this aspect from the manuscript.*

*We admit that the log-transformation was not consistently done in the reviewed version of the manuscript. We now corrected this and added the following explanation to the Material and Method section. "For the regression analyses, all soil chemical variables (element stocks, element concentrations, and element ratios) as well as the organic layer stock and the P concentration the parent material were transformed by calculating their natural logarithm (log-transformation) since they tended to be not normally distributed but were right-skewed. In addition, also the variable atmospheric N deposition was log-transform prior to regression analysis because it was not normally distributed." Further, we removed the results of the regression analysis from Fig. 1 which shows the linear models for the non-log transformed variables, for the sake of clarity. We like to show the non-transformed data in Fig.1 because the log transformed data is not necessarily helpful for understanding the relationships between the variables.*

SPECIFIC COMMENTS

*Line 13* – This study focused on soil organic matter C:N:P stoichiometry. Therefore not total P (TP), but organic P (OP) was reported on for the mineral soil. However, from the Methods section I understood that actually also TP and inorganic P (iP) were determined. If TP were used instead of OP, would conclusions remain similar? This can be relevant for comparison to other studies and datasets that determined TP, and sometimes not OP: e.g. Bo et al., 2020 – Forests, Hume et al., 2016 – Forest Ecology & Management, Kranabetter et al., 2020 – Biogeosciences. *Yes, for the mineral soil, we determine total P and organic P. However, in line 13 we explain the aim of the study (which is to investigate the ratios of C, N and P in the organic matter). It does not read well if we squeeze"total P" also into this sentence, and it does increase clarity, either.*

*Line 20* – "C:N ratios in the litter layer and mineral soil". I assume that "litter layer" should be "organic layer" since litter layers were excluded during sampling, according to *Line 100. Right, we corrected this.*

*Line 92* – Does "deciduous" refer to certain dominant species? Betula pendula? Does the dataset include some of the temperate Fagus sylvatica dominated forests in southern Sweden? *We added the information that deciduous forest refers to birch, aspen, beech or oak forest.*

*Line 110* – Mor and moder humus forms were selected, which excludes peatland. Were forests with a mull-type of humus, with no separate H layer but Ah layer also excluded? If so, was this for practical reasons, e.g. no real separate organic layer, and could that have biased any of the conclusions? *We excluded plots with the humus form mull because at these plots, the organic layer is not sampled separately from the mineral soil for the SFSI. We now added this information in the Material and method section.*

*Line 148* – Why were mixed forests pooled specifically with the spruce-dominated forests for the multiple regression analysis? *In the reviewed version of the manuscript we had tried to include the variable dominant tree species into the multiple regression analysis, by assigning numbers to the trees species. However, we now removed this aspect from the manuscript.*

*Line 149* – State that right-skewed variables were (natural?) log-transformed. This seems to be mostly done (as stated in the Results), but in a few graphs I still noticed at first sight + (right) skewed variables so that potentially not all model assumptions were met. See my comments there. *We now log-transformed all soil chemical data and the variable atmospheric N deposition before the regression analysis. Further, we removed the results of the regression analysis from Fig. 1 which shows the linear models for the non-log transformed variables for the sake of clarity.*

*Line 193* – There were only 10 data points for deciduous forests. To what extent are these representative for deciduous (Birch, Beech, ...) forests in the whole country with respect to the variables measured and conclusions we derive from the data? *We added the information that of the 10 deciduous forests, five were dominated by beech and oak forest.*

*Line 216* – I became a little confused here about which statements on C, N and P referred to concentrations, and which to stocks. Please explicitly state in this paragraph. *We now indicated this clearly throughout the paragraph.*

*Lines 262 and 270* – I agree that N2 fixation is ultimately responsible for increasing N stocks along the temperature gradient, but it is only part of the explanation for decreasing SOM C:N. What I miss here in this section is a reference to other microbial processes and plant-soil feedbacks. The latter you actually mention in the next two sections. Where it is warmer, not only N fixation rates are higher, but also N mineralization rates. Consequently more N becomes plant-available per unit of time, and plant tissues and litter will also have reduced C:N. The more N-enriched litter will then result in lower C:N organic matter. Some studies on plant-soil feedbacks and stoichiometry in boreal and global forests (not specifically along temperature gradients) are Hume et al., 2016 – Forest Ecology & Management; Shi et al., 2016 – Plant and Soil; Van Sundert et al., 2021 – European Journal of Forest Research. *We added the following lines to the manuscript. "Mineralization of N is also expected to increase with increasing temperature, and hence should counteract the accrual of N with increasing temperature. Our finding that the N stock of the organic layer increased with increasing MAT thus suggests that MAT has a larger positive effect on $N_2$ fixation than on net N mineralization."*

*Line 282* – "high N inputs can lead to P limitation in south Swedish forests": yes, and as you state in the first paragraph of the manuscript, P has mostly been neglected in boreal forests because of primarily widespread N limitation. But despite N limitation, considering P can be important for ecosystem function as shown by the first author in earlier studies. And some forests in Sweden, in the southwest and in valleys can be relatively N rich but poor in available P (Giesler et al., 1998 - Ecology). I suggest the authors to add a short section of 1-2 paragraphs at the end of the Discussion where avenues for future research are mentioned. See also my comment there. *We added another section about future research questions in the end of the Discussion (see above). Further, we included the mentioned reference in the Discussion*

*Lines 302 and 310* – "spruce tends to grow in more fertile soils than pine" and "first study to show that this difference in C:N (...) is also visible in the mineral subsoil, in a depth of 55-65 cm". This is indeed one of the first studies to show this result, +/- in contrast to for example Cools et al., 2014 at a Europe-wide scale. Differences in deeper soil C:N may occur because of belowground litter inputs and root activity, but isn't an alternative explanation at least as likely here: the large majority of these forests are plantations, and spruce is planted on already more fertile soil than pine. So perhaps deeper soil C:N was already lower for spruce than pine before planting, even if this occurred >= 60 years ago. Also, mention in the Methods section what percentage of the forests was natural vs planted. *We added the following information at the end of section 2.1: "Of the 309 sites, 119 sites had a stand age >120 years, and are*

*thus generally classified as old growth forests (not planted). Further, 25 sites were on formally set-aside land" Furthermore, we added the following sentence in the middle of section 4.2. "Yet, it has to be considered that we cannot clearly attribute the differences in stoichiometry to differences in vegetation since pine forests might have been established preferably on soils that already had nutrient poor SOM."*

*Line 340* – "litter layer" should be "organic layer" in this paragraph? If litter C:N:P stoichiometry was determined – which does not seem to be the case – some results could be added to the supplement to support discussions on plant-soil feedbacks elsewhere. *This was a mistake. We replaced "litter" by "organic".*

*Line 340* – References are missing in this short paragraph, and the sentences are a bit unclear. Please rephrase. For example, the second sentence appears to suggest that some bedrock can provide nitrogen (such bedrock does exist, e.g. Holloway & Dahlgren, 2002 – Global Biogeochemical Cycles), but this could be a grammatical issue. *We clarified the sentence and added a reference. It now reads as follows: "In addition, regarding the N concentration it could also be that fine-textured soils are commonly formed from nutrient-rich (potassium, phosphorus, magnesium, etc.) minerals which causes high plant productivity and $N_2$ fixation, resulting in higher N concentrations in fine-textured soils compared to coarse-textured soils (Clarholm and Skyllberg, 2013). This is likely also the reason for the higher N concentration of the organic layer in the fine-textured soils compared to the coarse-textured soils (Fig. 4f)."*

*Line 356* – I warmly recommend to add a 1-2 paragraph section 4.6 with "avenues for future research" or alike. Here, relevance and implications of the research can be further emphasized and discussed. For instance, can such newly available soil P and CNP stoichiometry data help in better quantifying regional and global-scale nutrient availability and limitation, defined and determined as in e.g. Van Sundert et al., 2019 – Global Change Biology? How can understanding of soil CNP gradients help in advancing global change research? ... Putting the research in such contexts can be of interest for the broader readership. *We added another section about future research questions in the end of the Discussion (see above).*

*Line 360* – Here, explicit mention is made to N:P AVAILABILITY. I agree with the statement, and some explanation of NP availabilities was written under section 4.2, but please explain a bit more in the newly suggested section 4.6. For example, while not the focus of this stoichiometry study, do you think that the iP and TP data could be useful for large-scale studies with more focus on N vs P availability? *We added another section about future research questions in the end of the Discussion (see above).Further, we also now included total P in the regression analysis.*

*Line 365* – Suggestion to not refer to numbered hypotheses ("in agreement with the third hypothesis") in the Conclusion. *The rationale behind this suggestion is not clear to us. We think that it is good to number the hypothesis because it allows it to refer clearly to one specific hypothesis.*

*Line 384* – I agree, the value of newly collected soil P data can not be overstated! *Yes!*

*Line 525* – Maybe showing correlation coefficients would be more useful than $R^2$ so that the reader sees the sign of change. *We added information on whether the variables were positively or negatively correlated in Table 1.*

*Table 1* – In the Abstract and text, the litter layer is mentioned about twice. Probably "organic layer" was meant there but if not, please add litter nutrient concentrations and ratios to the tables. *This was a mistake. We replaced "litter" by "organic".*

*Table 1* – Very good that "molar" was explicitly mentioned, as in terrestrial ecology/biogeochemistry mass-based ratios are also common (e.g. Hume et al., 2016 – Forest Ecology and Management; Manzoni et al., 2010 – Ecological Monographs; Vejre et al., 2003 – Soil Science Society of America). Was there

a specific reason to opt for molar ratios? *The reason to opt for molar ratios was that most papers about organic matter stoichiometry report element ratios on a molar basis. Hence, this facilitates comparison.*

*Table 2* – No interactions were tested among the explanatory variables. Why? Were there à priori reasons based on theory to exclude interactions from the analyses? Statistical reasons w.r.t. the dataset? *We now added the results of additional multiple regression analyses that consider interactions between the independent variables in Table 2.*

*Figure 2* – Some variables seem positively skewed, which can lead to violations of the homoscedasticity assumption of linear regression. Have you considered log-transforming, as you did elsewhere? Log-transforming may resolve heteroscedasticity for the soil variables (effectively + skewed?) but not growth (panel 2d ). For at least the latter I consider this acceptable because the regression line still passes through the middle of the point cloud, and tree stem growth was not the primary focus of the study. *We corrected this. For the regression analyses, all soil chemical variables (element stocks, element concentrations, and element ratios) as well as the organic layer stock and the P concentration the parent material were now transformed by calculating their natural logarithm (log-transformation) since they tended to be not normally distributed but were right-skewed.*

*Figure 2b* – assuming the relationship holds under log-transformation - Would you actually expect soil organic layer stocks to increase with MAT? Litter production (input) increases north-south, but also decomposition (output). Do the southern, warmer region data points represent more wet-soil (but not peaty) areas? Maybe this region has some drier sites without organic layer (e.g. mull humus) which were excluded from the dataset? ~bias? If so, this would not invalidate the main conclusions of the study but it may be important for interpretation of some results. *The relations holds under log transformation:*

[Figure]

*We added the following sentence to the Discussion "Similarly, the increase in the organic layer stock with increasing MAT (Fig. 2b) suggests that the MAT has a larger positive effect on plant productivity (Fig. 2d) than on decomposition. This in accordance with other studies showing stronger C accumulation in the organic layer in the South than in the North of Sweden (Akselsson et al., 2005)."*

*Figure S3* – [although correlations]: heteroscedasticity --> log-transformation of variables? *Now, in the revised manuscript, all soil chemical variables (element stocks, element concentrations, and element ratios) as well as the organic layer stock and the P concentration the parent material are transformed by calculating their natural logarithm (log-transformation) since they tended to be not normally distributed but were right-skewed. This applies also to Fig. S3.*

TECHNICAL CORRECTIONS

*Line 18* – "(...) MAT, almost twice as much as the organic layer stock increase along the MAT gradient." *We slightly changed the sentence.*

*Line 25* – "Further, we found that (...)" *We deleted the comma, as suggested.*

*Line 55* – "in Swedish forest soils" *We deleted one s.*

*Line 92* – "the following classes based on basal area:" *We replaced ; by :*

*Line 132* – "P in ignited and non-ignited samples" *That's what we wrote. This comment is not clear to us.*

*Line 147* – R version 4.1.1 is not from 2003. To get the most up to date citation for R you can use the citation() function of the statistical program. *It is from 2021. We corrected this.*

*Line 185* – "the C:N ratio of the mineral soil in PINE forests was on average 1.8 times higher" *Thanks! We corrected this.*

*Line 189* – "The C:P ratio of the organic layer in PINE forests was on average 1.3 times higher" *Thanks! We corrected this.*

*Line 127* – "Williams and Saunders (1956)". *Yes, we added a s.*

*Line 227* – "We analyzed the relationship between" (or relationships among) *We corrected this.*

*Line 270* – "the C:N ratio decreased": the relationship was significant, so I would remove the "tended to", also in light of earlier studies with more data points by the last author of this manuscript (Van Sundert et al., 2018 - Biogeosciences). *Done*

*Line 274* – "N:P ratio INCREASES with increasing MAT" *Thanks! We corrected this.*

*Line 279* – "foliage N:P ratio INCREASES with increasing N inputs" *Thanks! We corrected this.*

*Line 281* – "it has was suggested" Please correct. *Done*

*Line 305* – "some of the pine forests had" *We added a s.*

*Line 337* – "charged surfaces" *Done*

*Line 360* – "N:P ratio of the organic layer INCREASED substantially with increasing MAT, likely due to an INCREASE in the ratio of N:P availability with increasing MAT" *Thanks! We corrected this.*

*Line 362* – Please check grammar/structure of this sentence: "(...), as hypothesized, however, not (...)" *We corrected the grammar.*

*Line 565* – "temperature together with the N stock" *We added a s.*

*Table 1* – remove space: "0.13, p<0.001" *Done.*

*Figure 1* – "Map depicting mean annual temperature (MAT), ..." *Thanks! We corrected this.*

Reference

Akselsson, C., Berg, B., Meentemeyer, V., and Westling, O.: Carbon sequestration rates in organic layers of boreal and temperate forest soils—Sweden as a case study. Global Ecology and Biogeography, 14(1), 77-84, 2005.

---

## Author Comment (AC2)

**Answers to reviewer#2**

This study reports variation in soil C, N and OP stocks and stoichiometric ratios in organic layers and C, N and OP concentrations and ratios in mineral soil in a smaller subset of the large national Swedish forest soil inventory. It is a very comprehensive and spatially extensive data set, which is highly interesting by its inclusion of P as well as top- and subsoil layers of the mineral soil. I therefore recommend the paper for publication after attention to a number of issues mentioned below. *We thank the reviewer for positive evaluation of the manuscript and the detailed comments.*

The mixed forest stands need to be better described. Based on a figure heading it appears it was only pine-spruce mixtures, so how was coniferous- deciduous mixtures handled? Are they merged with some other category? *We added the information that deciduous forest refers to birch, aspen, beech or oak forest. Pine or spruce forests with a few deciduous trees are classified as mixed forests. However, the group mixed forest in this database is dominated mixed pine-spruce forests.*

The finding that subsoil stoichiometry was affected by tree species even in subsoil are interesting as we tend to believe and also often see that topsoil is mainly influenced by vegetation. As with all observational studies it is be good to discuss "hen and egg" questions like whether tree species affected subsoils or subsoils determined the natural regeneration or where specific species were selected for planting. This is also done in section 4.3. It is likely that texture differences are covarying with tree species, and therefore the subsoil C/N ratios are also lower in spruce than pine. Another possible problem with subsoils is that detection limits can give some bias. Was this an issue for e.g. N in 55-65 cm for which concentrations of particularly N but also C tends to get very low? It would be good to address this somewhere. *We added one sentence in the Discussion (lines 330-332) to address the "hen and egg" problem" in the following way. "Yet, it has to be considered that we cannot clearly attribute the differences in stoichiometry to differences in vegetation since pine forests might have been established preferably on soils that already had nutrient poor SOM."*

*The texture does hardly co-vary with tree species. This was (and is) clearly addressed in section 3.5, as follows:"Spruce forests tended to have a higher texture class, i.e., a finer soil texture than all other forests, while there was no substantial difference in texture between deciduous, mixed, and pine forest (Fig. S6c)." This is shown also in the Supplement in Fig. 6c (which was S5c in the reviewed version of the manuscript). The co-variance is (and was in the reviewed version of the manuscript) also discussed in section 4.3, as follows. "In addition, the relatively low C:N ratio in the mineral soil in spruce forests compared to pine forests might also partially result from the fact that the spruce forests tended to have a slightly finer texture (Supplement Fig. S6c), which is associated with lower C:N ratios".*

*We now added one sentence in the Results section to address the issue of the very low N contents in the following way. "It needs to be taken into account that there is some uncertainty regarding the samples with very low N and P concentrations since the determination of N and P contents in very nutrient poor samples is more strongly affected by sample inhomogeneity than the measurement of nutrient-rich samples."*

The effects of tree species and texture categories need to be more clearly communicated. None of the box plot figures enable separation of significant vs. non-significant mean values. The relative differences are mentioned but not whether such differences are significant. This needs to be improved. *We compare the tree species and texture categories in boxplots, in which not only the medians but also the arithmetic means are displayed, both by symbols and numbers (Figs. 3 and 4). In addition, the relative difference between texture classes is displayed in Fig. 4. The reviewer seems to request some information about which of these changes are important, meaningful or "real" and which not. However, the p value calculated in an ANOVA or the attribute "statistical significant" cannot provide this information (Amrhein et al., 2019; Wasserstein et al. 2019). Many statisticians caution today against the use of p values since random variation in the sampling of the same population can lead to a large disparities in p values and since p values are very often misused (Amrhein et al., 2019). We follow this advice here.*

The deciduous tree species group is a bit strange. There is a very high C/P ratio in min soil and highest OM stock in org layer? OM stocks in the organic layer are also extremely high in some cases. I cannot help speculate if a larger share of these sites are hydromorphic/peaty even though such sites were sought to be excluded? *Yes, the organic layer stock is largest in the deciduous forest. Yet, the C concentration in the mineral soil is in the same range as in the forests dominated by other tree species. We added a new figure to the Supplement to show this (Fig. S2c). As mentioned in the Material and Method section, we excluded peatlands. There is no stagnosol in this dataset and there are only three gleysols. However, none of them caries a deciduous forest (two are covered by pine forest and one by spruce forest). We now added this information to the Material and Method section. Based on this we see no reason to assume that hydromorphic features are the reason of the higher organic layer stock of the deciduous forests. The reason for the organic layer stocks of the deciduous forests might be related to the fact that the deciduous forests are concentrated at high MAT, where plant productivity is high.*

Lastly, there are a number of typos and reverted comparisons (higher/lower) to correct, and the finish in figures could be improved (please find details below). *We did so (see below).*

In conclusion, this is a solid manuscript based on a nicely planned study. The manuscript needs some attention to communicate results more clearly based on statistics, to clarify inconsistencies and consider some further methodological issues and aspects for the Discussion. *We thank the reviewer for positive evaluation of the manuscript and the detailed comments which helped us to improve the manuscript (see below).*

**Specific comments**

…adsorb particularly strongly…. *We added ly*

wood coring better than tree drilling? *We changed the wording.*

– what do the numbers in brackets stand for? *They refer to the code in the database. We now deleted them since they are not informative here.*

OK, I was initially surprised that the number of selected sites was so low, but based on these criteria it makes more sense. However, why was 60 years chosen rather than e.g. 30-40 years? Why only soils sampled 2013-2015? Could you have got substantially more sites and data from less rigid citeria, e.g. was the most limiting criterion the P analysis in parent material (which depth was this considered to be? It is definitely a nice dataset, but also quite small based on the huge forest area and high number of sampled plots. *Well, it can be debated whether 309 soil profiles is a small or a large number. Many papers in Biogeosciences are based on a much smaller number of soil profiles. We selected forests with a stand age >60 years in order to exclude a (recent) effect of clear-cutting and the associated soil preparation. Further, we only analyzed samples from the the current inventory (which started in 2013) for which the basic chemical data analysis had been completed (which at the time of manuscript preparation was 2018). The P concentration of the parent material was measured in a depth of 50 cm (which corresponds to the B or B/C horizon). We now added this depth to the method description.*

Was it concentrated nitric acid? For the mineral soil, was this truly "total" P – this would require HF as well or? *This procedure is widely accepted for the determination of total P, also in the mineral soil.*

Why was spruce and mixed stands combined? And were mixtures conifers+deciduous species or mixed spruce-pine? This is further in contrast to info in Fig. 3 where mixtures are shown separately, but are said in heading l. 573 to be only pine-spruce mixtures. *This is a misunderstanding. As indicated in this sentence, this referred only to the multiple regression analysis. In the reviewed version of the manuscript we had tried to include the variable dominant tree species into the multiple regression*

*analysis, by assigning numbers to the trees species. However, we now removed this aspect from the manuscript.*

144+ I miss information on the correlation analysis shown in Table 1 – is it simple correlations or linear regression as indicated by Fig. 1)? I also miss information on how the mineral soil layers were handled in analyses of C, N and P. *We added this information to the Material and Method section.*

Please indicate direction of correlations (pos/neg) in Table 1 as well. *We added signs in Table 1 to indicate positive and negative correlations.*

Why not show the correlation of N stock with log transformed N in Fig. 2? All other graphs match with Table 1. *We improved this aspect of the manuscript to separate more clearly the regression analysis (in Table 1 and 2) for which all soil variables have been log-transfomed now and the linear models shown in Fig. 1 (from which we now removed the results of the regression analysis).*

No need to repeat "molar" after M&M section. *Yes, in theory this is right, but since both mass and mol based ratios are used in this field of research it is better to keep this information in order to avoid confusion.*

– this should be the P stock increasing by a factor 2.3, right? *Yes. Thank you. We corrected this.*

It sounds strange that N:P ratio was negatively correlated with N deposition (given pattern in Fig. 2f). I would expect a positive correlation, also as N dep and MAT is closely intercorrelated. *We corrected this.*

It is unclear to me which layers of the mineral soil were analysed in Table 1? *We added in the caption that the data refer to the mineral soil in a depth of 0-10 cm.*

We need information on the direction (pos/neg) of these relationships for mineral soil even though they were weak. *We now added + and – in Table 1 to indicate this. In addition, we clarified this throughout the Results part.*

181+ In this section please indicate that the results are based on an ANOVA (?) and give the model already in the M&M section 2.3. *The reviewer seems to request some information about which of these changes are meaningful and which not. However, the p value calculated in an ANOVA or the attribute "statistical significant" cannot provide this information (Amrhein et al., 2019; Wasserstein et al. 2019). Many statisticians caution today against the use of p values since random variation in the sampling of the same population can lead to a large disparities in p values (Amrhein et al., 2019). We follow this advice here.*

I miss to see mineral soil C concentrations somewhere for the rtee species groups – can be added here or in the supplementary materials. *We added a new figure to the Supplement to show this (Fig. S2c).*

Please indicate significantly different species groups in Fig. 3. What does "inf" as the mean value indicate? *"inf" resulted from a mistake made when exporting the figure from R, which we now corrected by replacing inf by the correct numbers.*

I think you mean pine forests and not spruce forests here? *Yes, we corrected this.*

Please provide information whether C:N ratios were potentially biased because of low N concentrations (or C concentrations) that would be lower than analytical detection limits. *The number are above the quantification limit. We added the following lines in the Results. "It needs to be taken into account that there is some uncertainty regarding the samples with very low N and P concentrations*

*since the determination of N and P contents in very nutrient poor samples is more strongly affected by sample in homogeneities than the measurement of nutrient-rich samples."*

What happened to C:P ratios in 10-20 and 55-65 cm layers? *We only measured P in the organic layer and organic P and total P in the mineral soil in 0-20 cm depth.*

Here is an example why we need more specific evidence of effects: what does "…did not substantially differ…" mean exactly? *The reviewer seems to request some information about which of these changes are important or "real" and which not. We understand this request. However, the p value calculated in an ANOVA or the attribute "statistical significant" cannot provide this information (Amrhein et al., 2019; Wasserstein et al. 2019). Many statisticians caution today against the use of p values since random variation in the sampling of the same population can lead to a large disparities in p values (Amrhein et al., 2019). We follow this advice here.*

This is surprising – is the stock of OM in the organic layer really highest in deciduous stands? If so, I would expect a very unbalanced design for species along the climatic gradient or some interaction with drainage regime? Based on the high and much higher mean values cp to other species groups for deciduous species (149 t/ha) I wonder if there is still a substantial amount of hydromorphic (wet) soils with peaty topsoil included in the deciduous group? *None of the deciduous plots has a hydromorphic soil. As mentioned in the Material and Method section, we excluded peatlands. There is no stagnosol in this dataset and there are only three gleysols. However, none of them caries a deciduous forest (two are covered by pine forest and one by spruce forest). We now added this information to the Material and Method section. Based on this we see no reason to assume that hydromorphic features are the reason of the higher organic layer stock of the deciduous forests.*

197-207. This section is about texture effects. Why not go through soil chemistry variables as for tree species for consistence? Why not show the same CNP variables in Fig. 3 and 4 for better comparison? Of course fine to deviate if there is a point and you want to show what is significant for species and textre, respectively. But the strategy it is not clear. Why show both C:N ratio and N concentration in Fig. 4? Is it because you should have shown N stocks (kg/ha) instead as the text seems to indicate? *In general, we showed the most interesting findings in the figures. Further, when exploring relationship between SOM and texture (that might potentially stabilize SOM through sorption) it makes more sense to analyze the concentrations rather than the stocks.*

198-201 These sentences can be condensed – or the last one can be deleted. *We deleted one line.*

– you mean Fig. 4a-c? *Yes, we correct this.*

Sentence is wrong – should be opposite. Highest ratios in coarse-textured soils. *We corrected this sentence.*

205-207 This is not N stock in Fig. 4f, but N concentration…. What is right here? *We corrected this by replacing the word stock by concentration.*

210+ Nice to see the real data too for organic layers, and I think it would be fine to show just the non-linear version and the statistics based on logarithmic transformation. I believe, however, that the authors need to consider the organicdata here – it is said that peat soils were excluded, but these very influential sites with high organic layer mass have to be peat – organic layers well above 200 t/ha are somehow hydromorphic and must be >30 cm deep? *The sites with very high mass in the organic layer is actually associated with more fertile humus forms (moder type) with a significant mineral content, which contributes to the high mass. Looking at the carbon stock there are only 5 sites with >100 ton C / ha. One single site had a humus depth >30 cm (34 cm), but still the humus form was mor (which is acceptable under non hydro-morphic conditions).*

In this section, I would also mention the strong covariance between N deposition and latitude. *We added the following sentences here "Latitude and MAT were strongly negatively correlated ($R^2$=0.88). Further, the log-transformed N deposition rate was strongly negatively correlated with latitude ($R^2$=0.78), and strongly positively with MAT ($R^2$=0.82)."*

An increase from R2=0.16 to 0.20 is perhaps not so substantial? *We deleted the word substantially.*

I would start here with the correlation with latitude and then discuss if it is MAT or deposition which would be the main driver for organic layer N stock. *Yes.*

253-256 I do not understand this argument. Stem growth rate was clearly (as expected) positively related to MAT. I think gradients are somehow mixed up here. Please revise/reword. *Yes, it should have been "increased" not "decreased". We corrected this now.*

260+ But in southern Sweden, N deposition is generally larger than 4 kgN/ha as seen in Fig. 2c. But the low N dep/high MAT sites could see a contribution of N fixation to increase N stocks. Howeerv, I thin it would be valid to acknowledge that the spatial resolution of the Ndep map may not enable it to reflect local deposition conditions well enough. This could also be the reason for your weak correlation with N dep? *Yes. We added the following lines: "even if we take into account that the N deposition data is relatively uncertain and might not be fully representative of the accumulated differences in deposited N."*

What was the suggested mechanism in Högberg et al. – was N dep also not good as explanatory parameter? *The study also looks at a sequence at which N deposition and temperature are strongly correlated.*

mineral soil *Corrected*

Adsorption and protection against mineralization is a good point, but I think you need to discuss that fine-textured soils also have a much higher weathering capacity to release P? *We added a new figure to the supplement (now Fig. S3, see below).*

[Figure]

**Figure S3:** The total phosphorus concentration in the mineral soil in a depth of 0-10 cm depending on the soil texture (clay and fine silt n=11, coarse silt n=52, fine sand n=136, and sand n=110) in 309

Swedish forest soils with a stand age >60 years. The texture class called sand here encompasses sand and coarse sand. Black numbers give the median, red dots and red numbers depict the arithmetic mean.

*We added the following lines to section 3.3 "In contrast to OP, the total P concentration did not differ between soils with the texture class sand, find sand and coarse sand. Only soils with the texture clay showed elevated P concentrations compared to soils of the other three texture classes (Supplement Fig. S3). TP concentrations were higher in soils with the texture clay or fine silt than in soils with the texture sand or coarse sand by a factor of 1.9 (Supplement Fig. S3)."*
*Further, we added the following lines to section 4.4. "The fact that the total P concentration (Fig. S3 ) differed much less than the OP concentration (Fig. 4c) between the soils of different texture class supports our interpretation that the strong enrichment of OP in the fine-textured soils is mostly caused by rigid adsorption of OP compounds (that protects OP against decomposition) and much less by a higher P concentration or P availability in these soils."*

This effect of productivity on N stock in organic layers could also be in play to explain mineral soil N? *Yes. We added the following sentence. "In addition, regarding the N concentration it could also be that fine-textured soils are commonly formed from nutrient-rich (potassium, phosphorus, magnesium, etc.) minerals which causes high plant productivity and $N_2$ fixation, resulting in higher N concentrations in fine-textured soils compared to coarse-textured soils (Clarholm and Skyllberg, 2013)."*

Another reason could be that higher organic layer stocks are found at more (K and P) poor parent materials where decomposition rates are slow? *At least for P, we know that this is not true. We added the following sentence in the Results "The organic layer stock was not correlated with the P concentration of the parent material and only very weekly with the P concentration of the uppermost 10 cm of the mineral soil ($R^2$=0.01, p<0.05)."*

Table 1  Please indicate direction of correlations (pos/neg). *We now added this.*

Table 2 What is NTree? *We explained this now better in the Material and Method section (see above).*

Fig. 2 Heading l 565: N stock *Corrected.*

Fig. 3  Please indicate significant differences. What does "Inf" in red mean? Move x axis heading ("Tree Dom" – please write out) to x axes of Figs 3e and 3f. Heading l. 573: It is important to communicate the info about mixed stands earlier – and how mixed deciduous –coniferous stands were handled? *This was a mistake. We now replaced Inf by the correct number.*

Fig. 4 Again, we need indication of significant differences among texture classes. 4f: this is not N stock (l. 577) but N concentration….*Yes, we corrected the figure caption.*

**References**

Amrhein, V., Greenland, S., McShane, B. et al., (2019). Scientists rise up against statistical significance. *Nature 567*, 305-307.

Clarholm, M., & Skyllberg, U. (2013). Translocation of metals by trees and fungi regulates pH, soil organic matter turnover and nitrogen availability in acidic forest soils. *Soil Biology and Biochemistry*, *63*, 142-153.

Wasserstein, R. L., Schirm, A. L., & Lazar, N. A. (2019). Moving to a world beyond "p< 0.05". *The American Statistician*, *73*(sup1), 1-19.

---

## Author Comment (AC3)

**Answers to reviewer#3**

There is an unredeemed potential for interesting publications in the large databases of national forest/soil inventories. This is also the case for the Swedish Forest Soil Inventory, publications using these data are highly welcome. The present study focuses on carbon (C), nitrogen (N)and phosphorus (P) and the ratios C:N, C:P, and N:P in both organic and mineral soil layers which particularly the P and the ratio aspect are novel. After reading the paper, I am however, a bit puzzled. The manuscript/tittle and objective promises to explore relationship with climate, tree species and soil texture, and it delivers on showing these relationships, but the discussion offers very little or patchy interpretation. My greatest concern with regard to this manuscript is the lack of discussion of the results and relationship against important factors known to regulate the accumulation/loss of C from soil -the balance of input of organic material against decomposition. Instead the main part of the discussion is linked to changes in N and here the focus is on three hypotheses *" first, the higher N deposition towards the south/southwest of Sweden that enriches the organic layer in South/Southwest Sweden with N. Second, it could potentially be that the rate with which N is taken up by trees increases with decreasing MAT, leading to N depletion of the organic layer at sites with low MAT. Third, it might be that N2 fixation decreases strongly with decreasing MAT."* While I do not dispute that these could be relevant hypotheses, I find it strange that forest productivity and decomposition processes are not discussed more in full. The same criticism could be directed towards the discussion of the P, here I had expected that the regional distribution of P in parent material (geology) would have included, also a reflection regarding limits P nett-immobilisation and nett-mineralisation. The text is also drawn down by some unnecessary errors which could easily have been corrected before submission (see comments below). My conclusion is therefore that the manuscript is not ready for publishing, there are too many errors, and insufficient discussion of the results, see comments below for more specific reasons for my conclusion.

*In this study, we use data of the Swedish Forest Soil Inventory to elucidate the abiotic and biotic factors that control the ratios of carbon, nitrogen, and phosphorus in the organic matter of 309 forest soils. We concentrated the analysis on the effects of climate, dominant tree species, and soil texture on soil organic matter stoichiometry because we have data on these variables. In contrast, we do not have data on organic matter decomposition rates in these 309 forest soils or on net P immobilization and mineralization.*

*In the section cited by the reviewer, we discuss why the N stock of the organic layer changes more strongly than the organic layer stock with increasing mean annual temperature. We now added the following lines about N mineralization and decomposition to the manuscript "Mineralization of N is also expected to increase with increasing temperature, and hence should counteract the accrual of N with increasing temperature. Our finding that the N stock of the organic layer increased with increasing MAT thus suggests that MAT has a larger positive effect on $N_2$ fixation than on net N mineralization. Similarly, the increase in the organic layer stock with increasing MAT (Fig. 2b) suggests that MAT has a larger positive effect on plant productivity (Fig. 2d) than on decomposition."*

*We now also discuss decomposition in section 4.5 in the following way. "Further, it could be that the decomposition rate of the organic layer is positively related with the organic layer P concentration since it has been shown that P is the most limiting element for microbial activity during the first phase of decomposition of pine needles in Sweden (Staaf and Berg, 1982)."*

*We now included the P concentration of the parent material and the mineral topsoil (0-10 cm) in the regression analysis (see Table 1) showing that P was only very weakly related with latitude and mean annual temperature. We mentioned already in the previous version of the manuscript that the P concentration of the soil parent material did not differ much between the soils (see end of section 3.4). Hence we see no need for a detailed discussion of the "regional distribution of P in parent material" that the reviewer suggests.*

*We also added another figure showing the total P contents in soils of different texture class (see below).*

Does the paper address relevant scientific questions within the scope of BG? a. yes

2. Does the paper present novel concepts, ideas, tools, or data? a. Novel concepts: to some degree

b. Novel Ideas: the background for the first hypotheses is not clearly stated, but with better explanation it could be considered novel. I suggest that the authors should consider e.g. Sundqvist, M. K., Liu, Z. F., Giesler, R., Wardle, D. A., 2014, Plant and microbial responses to nitrogen and phosphorus addition across an elevational gradient in subarctic tundra, Ecology 95(7):1819-1835. Which presents N:P ratios in relationship with temperature gradient that coincide with the findings in the current study. *The background of hypothesis 1 was (and is) explained in lines 41-55. The study mentioned by the reviewer is not very well comparable since it is about a N and P addition experiment conducted in a tundra at different elevations.*

c. Novel tools: no

d. Novel data: yes

3. Are substantial conclusions reached? a. To some degree (hypothesis 1 is not treated satisfactory)

4. Are the scientific methods and assumptions valid and clearly outlined? a. More or less - with some concerns the most important listed here:

How are the ratios calculated, are they based on mass (concentration) or are they base on Molar ratio. The standard method in soil science is on mass - this is not that apparent for C:N ratio but will have large implications when calculating C:P and N:P ratios. Clarity in method is essential to know when comparing with other studies - though the relationship with MAT, tree species, texture will not be affected. Please check the C:P ratios given in figure 3e. *Most studies on stoichiometry in ecosystems report molar ratios. We now explain more explicitly how they were calculated in the Material and Method section. The ratios in Fig. 3e are correct.*

c. What is "organic lay stock" and how is it calculated

d. Several analyses that are referred to in the materials and methods are not used in the texts, Two of them I would hope you have considered and probably you have reason for not including - the pH and the P in the parent material?

5. Are the results sufficient to support the interpretations and conclusions? a. Yes and No, see specific comments below

6. Is the description of experiments and calculations sufficiently complete and precise to allow their reproduction by fellow scientists (traceability of results)? a. Some improvements needed - se comments below

7. Do the authors give proper credit to related work and clearly indicate their own new/original contribution? a. yes

8. Does the title clearly reflect the contents of the paper? a. yes

9. Does the abstract provide a concise and complete summary? a. The abstract only refers to relative distributions - I would recommend to include the range or levels of the ratios, concentrations, stocks.

10. Is the overall presentation well-structured and clear? a. Yes, in general

11. Is the language fluent and precise? a. Yes in general, see comments below

12. Are mathematical formulae, symbols, abbreviations, and units correctly defined and used? a. Formulas for calculation of stocks is missing and should be better explained, Also a specification of how the ratios were calculated should be included - from the figures I gather it is Molar ratio, but in the text it is not specified (both ratio and molar ratio is referred to) *In the previous version of the manuscript, we already explained how the stocks were calculated ("Organic layer element stocks were calculated by multiplying the organic layer stock with the respective element concentration") We now added the following information in the section 2.3: "All element ratios were calculated on a molar basis (mol mol$^{-1}$)."*

13. Should any parts of the paper (text, formulae, figures, tables) be clarified, reduced, combined, or eliminated? a. See comments below

14. Are the number and quality of references appropriate? a. Yes, sufficient

15. Is the amount and quality of supplementary material appropriate? a. yes

**Specific comments:**
L 15-16, repetition of >60 years- consider deleting one place or merging the two sentences
*We deleted "with a stand age >60 years" in line 15.*

L17, and L26 "organic layer stock" stock of what? should be it be "organic layer C stock"? or are we talking about layer thickness? volume? density? Mass?
*We are talking about the organic layer stock, which is the mass of the organic layer on an area basis (t ha$^{-1}$).*

Line 19: "went along with" - better "followed" *There is no temporal order involved here. Therefore we think that "went along with" is better than "followed".*

L24-25 "OP adsorbs very rigidly to mineral surfaces." Could this not also be an effect of higher P availability in fine textured soils? *Yes, in theory this could be. However this was not the case in the soils under study here. In order to show this we added a new figure to the supplement (now Fig. S3, see below), and we added the following lines to section 3.3.*

[Figure]

***Figure S3:** The total phosphorus concentration in the mineral soil in a depth of 0-10 cm depending on the soil texture (clay and fine silt n=11, coarse silt n=52, fine sand n=136, and sand n=110) in 309*

*Swedish forest soils with a stand age >60 years. The texture class called sand here encompasses sand and coarse sand. Black numbers give the median, red dots and red numbers depict the arithmetic mean.*

*"In contrast to OP, the total P concentration did not differ between soils with the texture class sand, find sand and coarse sand. Only soils with the texture clay had elevated P concentrations compared to soils of the other three texture classes (Supplement Fig. S3). TP concentrations were higher in soils with the texture clay or fine silt than in soils with the texture sand or coarse sand by a factor of 1.9 (Supplement Fig. S3)."*

Line26. "C and N concentration in the mineral soil" is a rather unprecise term as these concentrations normally are measured in several different horizons and are expected to differ significantly with depth and genetic horizon. If you were talking about stocks however, the correlations would appear more valid/informative. *We added "(0-10 cm depth)".*

L51: «most soils in Northern Europe are only between 9,000 and 14,000 years old". I believe you should then define Northern Europe - you need only to go to Denmark to find older soils. Perhaps I it is better to write "Sweden" or "Scandinavia" instead of "Northern Europe". *We replaced "Northern Europe" by "Scandinavia", as suggested.*

L50-55. It would be interesting to know at what limit, N:P ratio, you would expect P to become limiting. *We added "indicated by molar needle N:P ratio >12" in line 49.*

L78-80 "The soils are a representative subsample of Swedish forest soils with a stand age >60 years that were sampled for the Swedish Forest Soil Inventory. We selected sites with a stand age >60 years in order to minimize the effect of forest management on the results" will these cover both self-drained and poorly drained soils? *This is (and was) specified at the end of section 2.1.*

L104 - what are the numbers in the brackets referring to? - are they necessary? *They refer to the code in the database. We now removed them, as suggested by the reviewer.*

L 105-106: "and living and dead roots >1 mm diameter are removed." This is nearly an impossible task on dried soils - and according to personal experience would also introduce a bias - thin roots are easier to remove from coarse textured soils than from fine textured soils. I suggest that a better description would be " (where possible) living and dead roots were removed from the sample prior to sieving (<2mm) and homogenizing" of cause only if this is the case. *The separation was feasible since as much as 96% of the samples were on coarse silt or coarser soil texture. We now clarified that this sentence referred to the mineral soil.*

L109 -use "covered" instead of "vegetated" *We did as suggested.*

L113-114: "each three soils were classified as either leptosol, gleysol or cambisol, while one soil was an umbrisol" - Here I suggest a rephrasing: 3 were leptosol, 3 were gleysol, 3 were cambisol, and one was umbrisol *We did as suggested.*

L145: unclear "Organic layer element stocks were calculated by multiplying the organic layer stock with the respective element concentration." What is the "organic layer stock"? how is it calculated? *The organic layer stock is the mass of the organic layer on an area basis (t ha$^{-1}$). The determination of the organic layer stock is (and was) described in section 2 as follows "The organic layer is sampled volumetrically using a 10 cm diameter corer in a 3.14 m$^2$ subplot within each circular plot throughout the entire depth of the organic layer (up to 30 cm depth), excluding the litter layer. (...) the stock of the organic layer is calculated based on the weight of the fraction <2 mm of the organic layer"*

L168-169: There must be an error "The C:N ratio of the organic layer was negatively correlated with MAT (R2=0.08, Supplement Fig. S1) and with the log-transformed N deposition (R2=0.11) and latitude

(R2=0.08; Table1)." C:N ratios cannot be negatively correlated to latitude - they must be positive. It would have been better to have r values in table 1 showing either negative or positive correlations not just "$R^2$". *Yes, we corrected this by inserting "and positively with".*

L169-170: The molar C:N ratio is referred to for the first time -- are all ratios molar ratios? - not described in the M&M *We added the following sentence in section 2.3. "All element ratios were calculated on a molar basis (mol mol$^{-1}$)."*

L178-179: "Furthermore, C, N and OP in the mineral soil were only weakly correlated with latitude, MAT, and N deposition (Table 1)." Table may show correlations but no indication whether positive or negative. *We added this information in Table 1.*

L185." the C:N ratio of the mineral soil in spruce forests" should it not be pine? The same error applies for L189 with regard to C:P ratio *Yes, correct this now.*

L187: "Some pine forests had very high C:N and C:P ratios due to low N and P contents." In the materials and methods you should introduce some rules for discarding "unusual" ratios based on the detection limits of the elements. These ratios are "artifact" and any interpretation of these may be useless. *The samples are above the quantification threshold. We added the following lines in the Results "it needs to be taken into account that there is some uncertainty regarding the samples with very low N and P concentration since the determination of N and P contents in very nutrient poor samples is more affected by sample in homogeneities than the measurement."*

L192-194, "The organic layer stock differed also between forests with different dominant tree species (Supplement Fig. S2b). In deciduous forests, it was on average 2.2 times higher than in pine forests and 1.9 times higher than in spruce forests." This does not sound right - what is "organic layer stock"? *The organic layer stock is the mass of the organic layer on an area basis (t ha$^{-1}$), this was (and is) explained in section 2.1 (line 106 in the previous version of the manuscript).*

L202-204 : "The mean C:N ratio and C:OP ratio were 1.5 and 1.4 times higher in soils with the texture clay or fine silt than in soils with the texture sand or coarse sand (Fig. 4d and e)." Error - the opposite
□ The mean C:N ratio and C:OP ratio were 1.5 and 1.4 times **lower** in soils with the texture clay or fine silt than in soils with the texture sand or coarse sand (Fig. 4d and e). *Yes, we now corrected this.*

L244-246. Both "organic layer stock" and "organic matter stock" are used about the same parameter - stick to one (I believe it should be either better described and defined as well, see earlier comments) *In these lines, we refer to the organic layer stock as well as to the N stock of the organic layer. Both variables are well defined and it is clearly stated in the Material and Methods section how they were determined (see also comments above).*

L253 254: "the stem growth rate of trees decreased with MAT (Fig. 2d)" but figure 2 d show that the growth rate increases with MAT! *We corrected this.*

L279 "foliage N:P ratio decreases with increasing N inputs" should be increase? *Yes, we corrected this.*

L307 "miner" should be "mineral"? *Yes, we corrected this.*

L309-310 " However, this is the first study to show that this difference in the C:N ratio between forest dominated by different tree species is also visible in the mineral subsoil, in a depth of 55-65 cm, to our knowledge." But is this really confirmed by figure 3d? How have you handled the very low concentrations and detection limits? *The samples are above the quantification threshold. We added the following lines in the Results "It needs to be taken into account that there is some uncertainty regarding the samples with very low N and P concentration since the determination of N and P contents in very nutrient poor samples is more affected by sample in homogeneities than the measurement."*

L320 "changed" should be "charged", *We corrected this.*

L337 "changed" should be "charged" *We corrected this.*

L320-340 -4.4. Organic P and N contents are high in fine-textured soils, I find this paragraph too superficially discussed. *We added the following lines "The fact that the total P concentration differed much less than the OP concentration between the soils of different texture class (Fig. S3) supports our interpretation that the strong enrichment of OP in the fine-textured soils is mostly caused by rigid adsorption of OP compounds and much less by a higher P concentration of P availability in these soils." (Section 4.4). See also the new figure added (see above).*

L337: "Thus, the high capacity of N- and P-containing organic compounds to adsorb to changed surfaces in soils is likely the reason why N, and particularly OP has higher concentrations in the fine-textured soils compared to the coarse-textured soils than C." - this is not only due to sorption. These fine textured soils are more productive nutrient rich system - (P -rich included) - also the water holding capacity and pH increases fine textured soils. and what about regional distribution of P containing minerals? *See previous comment.*

L345 - 355 : 4.5 P and K contents are negatively related to the organic layer stock. The discussion in this paragraph is poor. These two elements behave very different in soil the potassium is a cation more or less only found in inorganic forms, while P mostly would be present as a anionic molecule and in organic form be part of organic molecule . K+ could easily be leached, even more so in acid conditions, while P-is retained and we would not expect any "fast leaching of P" I would find it highly unlikely that "P follows the same dynamic as K" *We revised this sentence, deleting "follows the same ...". We wrote: "The reasons for this might be that the decomposition rate of the organic layer is positively related with the organic layer P concentration. This is supported by a study showing that P is the most limiting element for microbial activity during the first phase of decomposition of pine needles in Sweden (Staaf and Berg, 1982). Further, it could be that there is a higher contribution of fine woody debris, which is P-poor (Spohn et al., 2020b) to the organic layer in forests with high organic layer stock."*

365 "the results show that the N:P ratio of the organic layer was most strongly related to MAT" showing a positive relationship, while the CN ratio showed a negative relationship with MAT. This confirms that the boreal forest are N deprived, any increase in N availability will increase forest productivity P is not the limiting factor *This argument this not clear to us.*

L570 figure 3e the molar C:P ratio of the organic layer is high - are you sure they are correct? *The C:P ratios of the organic layers are close to the mean C:P ratio of the organic layer in temperate forests (1411) found in a meta-analysis that synthesizes the results of many studies (Spohn, 2020 in GCB). Hence, they are not extraordinarily high.*

L570 figures b, c, d, and f, why do some of the means depict "Inf"? *This was a mistake. We corrected this and replaced Inf by the correct numbers.*

**References**

Staaf, H., & Berg, B. (1982). Accumulation and release of plant nutrients in decomposing Scots pine needle litter. Long-term decomposition in a Scots pine forest II. Canadian Journal of Botany, 60(8), 1561-1568.

Spohn, M. (2020b). Increasing the organic carbon stocks in mineral soils sequesters large amounts of phosphorus. Global Change Biology, 26(8), 4169-4177.

---

## Referee Report (RR1)

Title: Carbon, nitrogen, and phosphorus stoichiometry of organic matter in Swedish forest soils and its relationship with climate, tree species, and soil texture
Author(s): Marie Spohn and Johan Stendahl
bg-2021-346 Manuscript version 2MS

General comments:
The revised manuscript now appears in a much-improved state. Errors and mistakes have been corrected and unclarities have been clarified. The authors have followed recommendations from the different reviewers as conscientiously as could be expected. I will, however, maintain my initial remark that increase productivity due to climatic factors, growing season, the balance between decomposition and addition of organic matter is the main reason for positive correlation between N stocks and MAT, not increase in N2 fixation as the authors maintain. This set aside, I still recommend that the manuscript should be published. Some revision is needed before publishing, see comments below.

L203-204 "the C:N ratio of the mineral soil in spruce forests was on average
1.8 times higher than in deciduous forests and 1.2 times higher than in pine forests" Figure 3d does not show this it shows that pine>deciduous>spruce (spruce and pine should swap places in this sentence)

L208-209 "The C:P ratio of the organic layer in spruce forests was on average 1.3 times higher than in both deciduous and pine forests (Fig. 3e)." Again figure 3 e does not show this it shows pine>deciduous>spruce, - spruce and pine should swap places in this sentence as well.

L387-388 "thin organic layers that consist largely of relatively young organic matter have a higher K concentration than massive organic layers that mostly consist of old, K-poor organic matter." There is no reason to assume that a thin organic layer consists largely of relatively young organic matter. A thin organic layer may reflect 1) low input or 2) high decomposition rate and may span from low to high productive forest system - only in the high productive systems would a thin organic layer mostly consist of relatively young organic matter. If your data distinguishes between Oi, Oe or Oa then age could be more relevant to include in the discussion.

L390 (Figs. 4 and b) correct to (Figs 5a and b)

L390 -391"--agreement with the relationship reported by Stendahl et al. (2017) between the C stock of the organic layer and both its K and Mn concentration" The discussion of K could be more comprehensive - If it is in agreement with Stendahl et al 2017 why not also use their discussion. K can be related to better conditions for decomposition, or it could be related to water balance -and productivity? Trees in drought prone areas have lower K content than trees in areas with ample access to water - see e.g. Sardans J et al 2012.

L600 - 605, Table 1 and 2 I agree the p values should be included but I think the tables would be easier to read if only the p values that differ from the most prevalent are included.

---

## Author Response (AR2)

Dear Sara,

Thank you for handling the manuscript and for your letter!

We improved Figures 2 and 5 the way you suggested. We also improved the resolution of the figures in the Supplement. We will provide the figures for the manuscript as pdfs with a very high resolution, when given the opportunity.

Following reviewer#3's suggestion, we added another eight lines of text about decomposition in the organic layer to the Discussion on page 10. Further, we improved the structure of some sentences and corrected a few typos.

Kind regards,

Marie

**Letter from the editor:**

Dear authors,

I have now received the reports from the three referees who previously reviewed your manuscript. They are generally content with the changes that were made and I agree with them. A few suggestions for minor revisions were made that need to be taken into account before the manuscript can be accepted for publication. In addition, I list a few minor suggestions below.

- Figure 2: I think the text "increase from ...: factor x..." inside the figures could be removed. The necessary information is already provided in the figure and the equation, and is further detailed in the text.
- Figure 5: panels a and c have red lines, while panels b and d have black lines. Please harmonize this.
- In general, the quality of the figures is rather poor. Please improve this.

I look forward to receiving your revised manuscript.
Kind regards,
Sara

---

## Author Response (AR3)

Dear Sara,

We only received the pdf with the reviewer's comments now, but not with your previous message. Now that we have received the comments, we improved the manuscript accordingly. Please find our answers to all comments below.

In addition, we improved the sentence you mentioned, and carefully checked the whole manuscript again, and improved the structure of some sentences.

We also moved section 4.5 and the end of section 4.2 up to the beginning of the Discussion in order to have a straighter structure in the discussion. (These sections are now marked as new text in the document with tracked changes, although they have only been moved up.)

We thank you and the reviewer very much for the constructive comments!
Kind regards,

Marie

Title: Carbon, nitrogen, and phosphorus stoichiometry of organic matter in Swedish forest soils and its relationship with climate, tree species, and soil texture
Author(s): Marie Spohn and Johan Stendahl
bg-2021-346 Manuscript version 2MS

General comments:
The revised manuscript now appears in a much-improved state. Errors and mistakes have been corrected and unclarities have been clarified. The authors have followed recommendations from the different reviewers as conscientiously as could be expected. I will, however, maintain my initial remark that increase productivity due to climatic factors, growing season, the balance between decomposition and addition of organic matter is the main reason for positive correlation between N stocks and MAT, not increase in N2 fixation as the authors maintain. This set aside, I still recommend that the manuscript should be published. Some revision is needed before publishing, see comments below.
We thank the reviewer very much for the positive evaluation of the revision. The balance between decomposition and addition of organic matter is now mentioned in the beginning of the Discussion **(lines 276-278**), in the following way "*The positive relationship between the organic layer stock and MAT (Fig. 2b) suggests that plant productivity (Fig. 2d) increases more strongly than decomposition from North to South Sweden, which is likely due to differences in the temperature regime among the plots.*"
We also added some lines about the role of decomposition for the latitudinal differences in the N stock in the Discussion on page 10 (**lines 317-320**), which read as follows. "*Third, it could potentially be that the net N mineralization rate is negatively related with MAT, and thus is lower in the South where the C:N ratios are comparatively low (Fig. S1) than in the North of Sweden. However, the net N mineralization rate in the organic layer of temperate and boreal forests is usually negatively related with the C:N ratio (Parton et al., 2007; Moore et al., 2011; Heuck and Spohn, 2016) which rather suggests a high N mineralization rate at sites with high MAT and low C:N ratio, as also discussed recently by Högberg et al. (2021).*"

L203-204 "the C:N ratio of the mineral soil in spruce forests was on average 1.8 times higher than in deciduous forests and 1.2 times higher than in pine forests" Figure 3d does not show this it shows that pine>deciduous>spruce (spruce and pine should swap places in this sentence).
Thank you! We corrected this.

L208-209 "The C:P ratio of the organic layer in spruce forests was on average 1.3 times higher than in both deciduous and pine forests (Fig. 3e)." Again figure 3 e does not show this it shows pine>deciduous>spruce, - spruce and pine should swap places in this sentence as well.
Thank you! We corrected this.

L387-388 "thin organic layers that consist largely of relatively young organic matter have a higher K concentration than massive organic layers that mostly consist of old, K-poor organic matter." There is no reason to assume that a thin organic layer consists largely of relatively young organic matter. A thin organic layer may reflect 1) low input or 2) high decomposition rate and may span from low to high productive forest system - only in the high productive systems would a thin organic layer mostly consist of relatively young organic matter. If your data distinguishes between Oi, Oe or Oa then age could be more relevant to include in the discussion.
We agree that thin organic layers do not necessarily have to consist of young organic matter, and we removed the part of the sentence that referred to age. The sentence now reads as follows "*The reason for this negative correlation could be faster leaching of K from thinner than from more massive organic layers (Osono and Takeda, 2004; Schlesinger et al., 2021).*"

L390 (Figs. 4 and b) correct to (Figs 5a and b) We corrected this.

L390 -391"--agreement with the relationship reported by Stendahl et al. (2017) between the C stock of the organic layer and both its K and Mn concentration" The discussion of K could be more comprehensive - If it is in agreement with Stendahl et al 2017 why not also use their discussion. K can be related to better conditions for decomposition, or it could be related to water balance –and productivity? Trees in drought prone areas have lower K content than trees in areas with ample access to water - see e.g. Sardans J et al 2012. Stendahl et al. (2017) say that K limitation of decomposition "seems unlikely for a majority of the data". We added the following sentence to our manuscript "*K has also been reported to be related to the decomposition rate of needles in Scots pine forests, yet the underlying mechanisms are not well understood (Laskowski et al., 1995; Stendahl et al., 2017).*" The mentioned paper by Sardans et al. is about Mediterranean forests, and we feel that it would be quite a stretch to claim that the pattern observed there with respect to aridity can explain our findings.

L600 - 605, Table 1 and 2 I agree the p values should be included but I think the tables would be easier to read if only the p values that differ from the most prevalent are included.
We would like to keep the p values in the table for clarity; also because we only give the p value in the text if it differs from p<0.001 (as also stated in Material and Methods).